

# Logarithmic CFT at generic central charge: from Liouville theory to the $Q$-state Potts model

Rongvoram Nivesvivat[*] and Sylvain Ribault[†]

Université Paris-Saclay, CNRS, CEA, Institut de physique théorique

[*] rongvoram.nivesvivat@ipht.fr, [†] sylvain.ribault@ipht.fr

## Abstract

Using derivatives of primary fields (null or not) with respect to the conformal dimension, we build infinite families of non-trivial logarithmic representations of the conformal algebra at generic central charge, with Jordan blocks of dimension 2 or 3. Each representation comes with one free parameter, which takes fixed values under assumptions on the existence of degenerate fields. This parameter can be viewed as a simpler, normalization-independent redefinition of the logarithmic coupling. We compute the corresponding non-chiral conformal blocks, and show that they appear in limits of Liouville theory four-point functions.

As an application, we describe the logarithmic structures of the critical two-dimensional $O(n)$ and $Q$-state Potts models at generic central charge. The validity of our description is demonstrated by semi-analytically bootstrapping four-point connectivities in the $Q$-state Potts model to arbitrary precision. Moreover, we provide numerical evidence for the Delfino–Viti conjecture for the three-point connectivity. Our results hold for generic values of $Q$ in the complex plane and beyond.



# 1 Introduction

Following the lead of its non-logarithmic counterpart, the study of two-dimensional logarithmic CFT has led to a good understanding of chiral structures, in particular logarithmic representations and their fusion rules [1]. However, powerful results on chiral logarithmic CFT are not always easy to translate into a good understanding of bulk logarithmic CFT. Bulk CFT involves coupling left- and right-moving chiral structures, and this coupling can be nontrivial [2]. Then is it possible to study the bulk theory without starting from the chiral theory? We propose a positive answer to this question, in the case of CFTs based on the Virasoro algebra at generic central charge. This case is relatively simple algebraically, and it is motivated by the $O(n)$ model and the $Q$-state Potts model. These models are known to be logarithmic at certain rational central charges, and have long been suspected of being logarithmic at generic central charge.

**Logarithmic fields as derivatives**

Of course, at any central charge and even in any dimension, we can obtain simple logarithmic fields by taking derivatives of primary fields with respect to the conformal dimension [3,4]. We will focus on the next simplest case, and build logarithmic fields from primary fields that have null vectors. Our main technical tool will still be derivatives with respect to the conformal dimension. Actually, we can not only differentiate, but also apply any linear operation: in particular, following [5], we will linearly combine primary fields with descendant fields.

This approach is particularly effective for computing correlation functions and conformal blocks. Four-point conformal blocks need not be computed by summing over states in logarithmic representations: they can be obtained by combining standard conformal blocks for Verma modules. This will allow us to determine four-point conformal blocks for representations whose logarithmic features appear at arbitarily high level, and ultimately to compute connectivities in the Potts model to arbitrary precision.

**Constraints from degenerate fields**

Constructing logarithmic representations from derivative fields says nothing on the possible appearance of these representations in particular CFTs. However, the existence of degenerate fields in a CFT can imply the existence of logarithmic representations, and determine their structures [6]. In the cases of the $O(n)$ model and of the $Q$-state Potts model, this idea was recently used in [7], although it was only worked out for a subset of logarithmic representations. We will follow this idea more systematically, and write a conjecture for the structures of all logarithmic representations in these models at generic central charge. (See Section 4.)

The determination of the structures of logarithmic representations includes the determination of their parameters, sometimes called logarithmic couplings. We will find explicit expressions for these parameters, which not only hold at generic central charge, but also seem to make sense at rational central charge, as we will show in examples. (See Section 3.1.)

**Bootstrapping connectivities**

Our logarithmic structures may seem somewhat speculative, as they rely on using derivative fields in theories with discrete spectrums, and assuming the existence of degenerate fields. In order to validate our techniques and assumptions, we will first show that our constructions make sense in the context of Liouville theory, whose continuous spectrum allows us to differentiate physical fields, and where degenerate fields are known to exist. (See Section 3.3.) Then we will bootstrap four-point connectivities in the *Q*-state Potts model. A recent attempt at bootstrapping these connectivities was broadly successful [8], although its accuracy was limited by the presence of unknown logarithmic contributions. We will overcome this limitation, and solve crossing symmetry equations to a high accuracy. (See Section 5.) Our Python code for computing four-point connectivities is available at GitLab [9].

According to forthcoming work by Grans-Samuelsson et al, predictions from the lattice discretization of the *Q*-state Potts model also seem to converge towards the same logarithmic structures [10].

## 2 Logarithmic fields as derivatives of primary fields

Primary fields play a central role in conformal field theory, because they generate the simplest and most common representations of the conformal algebra: Verma modules and quotients thereof. In such representations, the dilation generator is diagonalizable. We will now investigate logarithmic representations, where by definition the dilation generator is not diagonalizable.

We will build logarithmic representations from derivatives of primary fields with respect to the conformal dimension. This approach is technically convenient, because it allows us to easily compute correlation functions, conformal blocks and operator product expansions: since these objects are solutions of linear Ward identities, their derivatives are solutions as well. In two dimensions, this approach has the added advantage of guaranteeing the single-valuedness of correlation functions: if we separately studied left- and right-moving fields and representations, we would face the additional problem of combining them in a single-valued way, in other words of building bulk CFT from chiral CFT.

From the derivative of a primary field, we build a logarithmic representation whose structure depends on the primary field's properties. We will first focus on the simplest case of a non-degenerate primary field, i.e. a primary field that generates an irreducible Verma module. This case is well-known [3], and has even been investigated in higher-dimensional CFT [4]. We will then move on to the case where a null vector (= a singular vector) is present. This case has already been investigated in two dimensions [11] and higher dimensions [12]. Things cannot get more complicated at generic central charge, as representations with several null vectors only appear at rational central charge.

The novelty in our approach therefore comes neither from the derivative field techniques, nor from the algebraic structures of logarithmic representations, but from the combination of these two elements. Namely, we will use derivative field techniques for determining the parameters of logarithmic representations, under assumptions on the existence of degenerate fields. Moreover, our definition of these parameters will allow us to bypass the awfully pedestrian calculations that sometimes appear in the literature, and to determine these parameters

in the presence of null fields with arbitrary levels.

## 2.1 Derivatives of primary fields

**Jordan blocks from derivatives**

Let us start with a primary field $V_\Delta$ with the conformal dimension $\Delta$. By definition, this is a field on which the dilation generator $L_0$ and annihilation modes $L_{m>0}$ act as

$$L_0 V_\Delta = \Delta V_\Delta \,, \tag{2.1}$$

$$L_{m>0} V_\Delta = 0 \,. \tag{2.2}$$

Our notations for the symmetry generators $L_m$ come from the Virasoro algebra which is relevant to the two-dimensional case. Let us take $\Delta$-derivatives of these equations, while considering the operators $L_0, L_{m>0}$ as $\Delta$-independent. Let

$$\hat{V}_\Delta = \left[ \frac{1}{n!} V_\Delta^{(n)}, \cdots, \frac{1}{2} V_\Delta'', V_\Delta', V_\Delta \right]^T \tag{2.3}$$

be the vector of derivatives of $V_\Delta$ up to $V_\Delta^{(n)}$. We then have

$$\hat{L}_0 \hat{V}_\Delta = \hat{\Delta} \hat{V}_\Delta \,, \tag{2.4}$$

$$\hat{L}_{m>0} \hat{V}_\Delta = 0 \,, \tag{2.5}$$

where $\hat{L}_m$ is the diagonal matrix with $L_m$ on the diagonal, and we introduce

$$\hat{\Delta} = \begin{bmatrix} \Delta & 1 & 0 & \cdots & 0 \\ 0 & \Delta & 1 & \ddots & 0 \\ 0 & 0 & \Delta & \ddots & 0 \\ \vdots & \ddots & \ddots & \ddots & 1 \\ 0 & 0 & 0 & 0 & \Delta \end{bmatrix} \,. \tag{2.6}$$

This describes a Jordan block of dimension $n + 1$.

The primary field $V_\Delta$ is defined up to a $\Delta$-dependent normalization factor, and this leads to ambiguities in the definition of $V_\Delta'$ too:

$$V_\Delta \to \lambda(\Delta) V_\Delta \quad \implies \quad V_\Delta' \to \lambda(\Delta) V_\Delta' + \lambda'(\Delta) V_\Delta \,. \tag{2.7}$$

More generally, in the module generated by $V_\Delta^{(n)}$, this change of normalization leads to a change of bases that preserves the action of $L_0$. A change of bases is not a change of structure, and the module has no free parameters.

**Two-dimensional case: diagonal primary fields**

In two dimensions, the conformal algebra factorizes into a product of two Virasoro algebras, called left-moving and right-moving. A field that is primary for both Virasoro algebras is characterized by a left-moving and a right-moving conformal dimensions called $\Delta$ and $\bar{\Delta}$. Single-valuedness of correlation functions on the sphere requires $\Delta - \bar{\Delta} \in \frac{1}{2}\mathbb{Z}$. (See [13] for a review.) The simplest way to fulfil this constraint is to consider diagonal fields, i.e. fields with $\Delta = \bar{\Delta}$.

There are known conformal field theories, such as Liouville theory, that involve diagonal primary fields whose dimensions can vary continuously. This encourages us to consider derivatives of diagonal primary fields. On the other hand, we do not know of any theory that would

involve non-diagonal primary fields with continuously varying dimensions. We will therefore refrain from building logarithmic representations from derivatives of non-diagonal primary fields.

From now on, $V_\Delta$ will denote a diagonal primary field whose left and right dimensions are both $\Delta$. Then the derivative field $V'_\Delta$ obeys

$$L_0 V'_\Delta = \bar{L}_0 V'_\Delta = \Delta V'_\Delta + V_\Delta \ . \tag{2.8}$$

It follows that the representation generated by $V'_\Delta$ cannot be factorizable. To prove this, let us concentrate on the two-dimensional subspace $\mathrm{Span}(V_\Delta, V'_\Delta)$, which we view as a representation of the subalgebra $\mathrm{Span}(L_0, \bar{L}_0)$. If this subspace could be factorized as a tensor product of representations of $L_0$ and $\bar{L}_0$, one of the two factors would have dimension one, and $V'_\Delta$ would be an eigenvector of either $L_0$ or $\bar{L}_0$.

## 2.2 Derivatives of null fields

**Null fields**

Let us rewrite the central charge $c$ in terms of a coupling constant $\beta$, and the conformal dimension $\Delta$ in terms of a momentum $P$:

$$c = 1 - 6\left(\beta - \frac{1}{\beta}\right)^2 \quad , \quad \Delta = \frac{c-1}{24} + P^2 \ . \tag{2.9}$$

The condition that the Verma module $\mathcal{V}_\Delta$ with the conformal dimension $\Delta$ has a null vector is

$$\mathcal{V}_\Delta \text{ has a null vector} \iff \Delta \in \left\{\Delta_{(r,s)}\right\}_{r,s \in \mathbb{N}^*} \ , \tag{2.10}$$

where the degenerate dimensions $\Delta_{(r,s)}$ correspond to the momentums

$$P_{(r,s)} = \frac{1}{2}\left(\beta r - \beta^{-1} s\right) \ . \tag{2.11}$$

We will use the notation $V_{(r,s)} = V_{\Delta_{(r,s)}}$ for a diagonal primary field of dimension $\Delta_{(r,s)}$. Let us write the null fields in the corresponding Verma module as $\mathcal{L}V_{(r,s)} = \mathcal{L}_{(r,s)}V_{(r,s)}$, where $\mathcal{L}_{(r,s)}$ is a creation operator. For example, $\mathcal{L}_{(1,1)} = L_{-1}$ and $\mathcal{L}_{(2,1)} = L_{-1}^2 - \beta^2 L_{-2}$. The same primary field also has a right-moving null descendant $\bar{\mathcal{L}}V_{(r,s)}$. Then $\mathcal{L}\bar{\mathcal{L}}V_{(r,s)}$ is a diagonal primary field of dimension $\Delta_{(r,-s)}$, and we make the identification

$$V_{(r,-s)} = \mathcal{L}\bar{\mathcal{L}}V_{(r,s)} \ . \tag{2.12}$$

Null fields can consistently be set to zero, and they do vanish in CFTs such as minimal models. However they do not have to vanish, and we assume that our null fields do not vanish, in particular $\mathcal{L}\bar{\mathcal{L}}V_{(r,s)} \neq 0$. We plot the four primary fields $V_{(r,s)}, \mathcal{L}V_{(r,s)}, \bar{\mathcal{L}}V_{(r,s)}, V_{(r,-s)}$ according to

their left and right conformal dimensions:

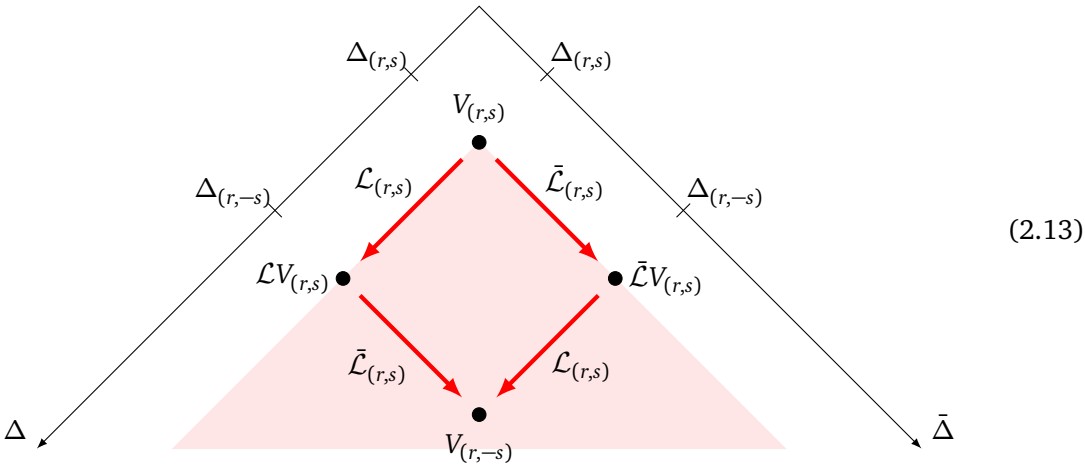

(2.13)

**Combinations of derivatives**

We have considered a representation that contains two diagonal primary fields, namely $V_{(r,s)}$ and $V_{(r,-s)}$. From their derivatives $V'_{(r,s)}$ or $V'_{(r,-s)}$, we could generate logarithmic representations that would not differ much from the representations from Section 2.1. To build something new, we introduce the linear combination

$$W^{\kappa}_{(r,s)} = (1-\kappa)V'_{(r,-s)} + \kappa \mathcal{L}\bar{\mathcal{L}}V'_{(r,s)} .$$

(2.14)

We call $\mathcal{W}^{\kappa}_{(r,s)}$ the representation of the product of two Virasoro algebras that is generated by the field $W^{\kappa}_{(r,s)}$. We have fixed the sum of the coefficients to one by a choice of normalization, but $\kappa$ is a normalization-independent parameter of the representation.

Let us investigate the properties of the representation $\mathcal{W}^{\kappa}_{(r,s)}$. We first compute

$$\left(L_0 - \Delta_{(r,-s)}\right)W^{\kappa}_{(r,s)} = \left(\bar{L}_0 - \Delta_{(r,-s)}\right)W^{\kappa}_{(r,s)} = V_{(r,-s)} ,$$

(2.15)

where we used the equations (2.8) and (2.12). This shows that the representation $\mathcal{W}^{\kappa}_{(r,s)}$ is logarithmic for any finite value of $\kappa$. On the other hand, $W^{\infty}_{(r,s)}$ is an eigenvector of $L_0$, and generates a non-logarithmic representation.

For any creation operator $\mathcal{L}$, annihilation operator $\mathcal{A}$, and conformal dimension $\Delta$, let us define a $\Delta$-dependent creation operator $[[\mathcal{A}, \mathcal{L}]](\Delta)$ by

$$\mathcal{A}\mathcal{L}V_{\Delta} = [[\mathcal{A}, \mathcal{L}]](\Delta)V_{\Delta} .$$

(2.16)

For example, $[[L_1, L_{-1}^2]](\Delta) = (4\Delta + 2)L_{-1}$. Then $[[\mathcal{A}, \mathcal{L}]](\Delta)$ depends polynomially on $\Delta$. Let us now focus on the case $\mathcal{L} = \mathcal{L}_{(r,s)}$. Since $\mathcal{A}\mathcal{L}V_{(r,s)} = 0$, we have

$$[[\mathcal{A}, \mathcal{L}_{(r,s)}]](\Delta_{(r,s)}) = 0 .$$

(2.17)

Differentiating with respect to $\Delta$, we then find

$$\mathcal{A}\mathcal{L}V'_{(r,s)} = [[\mathcal{A}, \mathcal{L}_{(r,s)}]]'(\Delta_{(r,s)})V_{(r,s)} .$$

(2.18)

Therefore, the action of an annihilation operator on the field $W^{\kappa}_{(r,s)}$ is

$$\mathcal{A}W^{\kappa}_{(r,s)} = \kappa[[\mathcal{A}, \mathcal{L}_{(r,s)}]]'(\Delta_{(r,s)})\bar{\mathcal{L}}V_{(r,s)} .$$

(2.19)

Let us consider the special case where $\mathcal{A}$ is an annihilation operator of degree $rs$, which we now denote $\mathcal{D}$: for example, $\mathcal{D} = L_1^{rs}$. The operator $[[\mathcal{D}, \mathcal{L}_{(r,s)}]](\Delta)$ is actually proportional to the identity, and we treat it as a number, not an operator. As a polynomial in $\Delta$, it has a zero at $\Delta = \Delta_{(r,s)}$, and this zero is simple if the central charge is generic. Applying $\mathcal{D}$ to the field $W_{(r,s)}^{\kappa}$, we climb back to the primary fields $\mathcal{L}V_{(r,s)}$ and $\bar{\mathcal{L}}V_{(r,s)}$. For simplicity, we now assume that $\mathcal{D}$ is normalized such that

$$[[\mathcal{D}, \mathcal{L}_{(r,s)}]]^{'}(\Delta_{(r,s)}) = 1 . \tag{2.20}$$

We then have

$$\mathcal{D}W_{(r,s)}^{\kappa} = \kappa\bar{\mathcal{L}}V_{(r,s)} \quad , \quad \bar{\mathcal{D}}W_{(r,s)}^{\kappa} = \kappa\mathcal{L}V_{(r,s)} . \tag{2.21}$$

We deduce closed equations for $W_{(r,s)}^{\kappa}$,

$$\boxed{\mathcal{L}_{(r,s)}\mathcal{D}W_{(r,s)}^{\kappa} = \bar{\mathcal{L}}_{(r,s)}\bar{\mathcal{D}}W_{(r,s)}^{\kappa} = \kappa\left(L_0 - \Delta_{(r,-s)}\right)W_{(r,s)}^{\kappa} = \kappa\left(\bar{L}_0 - \Delta_{(r,-s)}\right)W_{(r,s)}^{\kappa}} . \tag{2.22}$$

We can also rewrite Eq. (2.19) as a closed equation,

$$\boxed{\mathcal{A}W_{(r,s)}^{\kappa} = [[\mathcal{A}, \mathcal{L}_{(r,s)}]]^{'}(\Delta_{(r,s)})\mathcal{D}W_{(r,s)}^{\kappa}} . \tag{2.23}$$

Let us plot the resulting representation $\mathcal{W}_{(r,s)}^{\kappa}$, with black dots for primary fields, and a blue circle for $W_{(r,s)}^{\kappa}$:

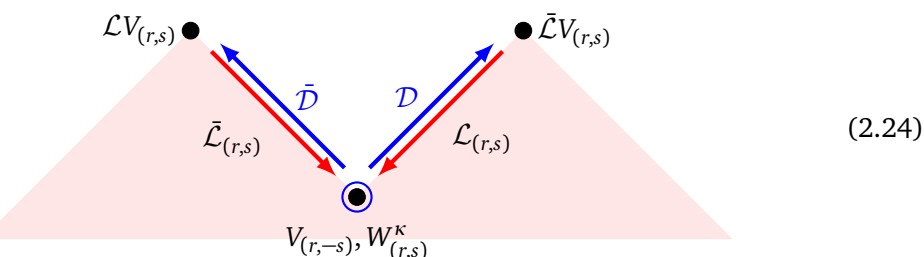

$$\tag{2.24}$$

Although the diagonal primary field $V_{(r,s)}$ does not belong to $\mathcal{W}_{(r,s)}^{\kappa}$, we use the notations $\mathcal{L}V_{(r,s)}, \bar{\mathcal{L}}V_{(r,s)}$ for two non-diagonal primary fields that do. The subrepresentation that they generate can be built by joining the Verma modules generated by each field along their common submodule, which is generated by the diagonal primary field $V_{(r,-s)}$:

$$\frac{\mathcal{V}_{(r,s)} \otimes \bar{\mathcal{V}}_{(r,-s)} \oplus \mathcal{V}_{(r,-s)} \otimes \bar{\mathcal{V}}_{(r,s)}}{\mathcal{V}_{(r,-s)} \otimes \bar{\mathcal{V}}_{(r,-s)}} \subset \mathcal{W}_{(r,s)}^{\kappa} . \tag{2.25}$$

Finally, let us show that the parameter $\kappa$ uniquely characterizes the representation $\mathcal{W}_{(r,s)}^{\kappa}$, and is normalization-independent. Given the representation $\mathcal{W}_{(r,s)}^{\kappa}$, we first have to find the field $W_{(r,s)}^{\kappa}$. From its definition (2.14) as a derivative field, we already know that this field is not quite unique, as we can perform a change of field normalization (2.7) that is compatible with Eq. (2.12), i.e.

$$\lambda(\Delta_{(r,s)}) = \lambda(\Delta_{(r,-s)}) . \tag{2.26}$$

Under such a change of field normalization, the field $W_{(r,s)}^{\kappa}$ behaves as

$$W_{(r,s)}^{\kappa} \rightarrow \lambda(\Delta_{(r,s)})W_{(r,s)}^{\kappa} + \left[(1-\kappa)\lambda'(\Delta_{(r,-s)}) + \kappa\lambda'(\Delta_{(r,s)})\right]V_{(r,-s)} , \tag{2.27}$$

which amounts to a change of bases of the representation $\mathcal{W}_{(r,s)}^{\kappa}$. Let us argue that the algebraic definition (2.22), (2.23) of the field $W_{(r,s)}^{\kappa}$ only allows this ambiguity and nothing more. Since by definition $W_{(r,s)}^{\kappa}$ generates the representation $\mathcal{W}_{(r,s)}^{\kappa}$, and since $L_{n\geq 0} W_{(r,s)}^{\kappa}$ is an eigenvector of $L_0, \bar{L}_0$, the only fields that can be added to $W_{(r,s)}^{\kappa}$ are $(L_0, \bar{L}_0)$-eigenvectors with eigenvalues $(\Delta_{(r,-s)}, \Delta_{(r,-s)})$. Moreover, these eigenvectors can always be written as $\mathcal{M}\mathcal{D}W_{(r,s)}^{\kappa}$ for some creation operator $\mathcal{M}$ of level $rs$. In order for Eq. (2.23) to be unchanged under $W_{(r,s)}^{\kappa} \to W_{(r,s)}^{\kappa} + \mathcal{M}\mathcal{D}W_{(r,s)}^{\kappa}$, we would need

$$\forall \mathcal{A}, \quad \Big([[\mathcal{D}, \mathcal{L}_{(r,s)}]]'[[\mathcal{A}, \mathcal{M}]] - [[\mathcal{D}, \mathcal{M}]][[\mathcal{A}, \mathcal{L}_{(r,s)}]]'\Big)(\Delta_{(r,s)}) = 0. \tag{2.28}$$

This system of linear equations has the obvious solution $\mathcal{M} = \mathcal{L}_{(r,s)}$, which corresponds to the ambiguity (2.27). This is the only solution if the central charge is generic, as we found by numerical explorations for $rs \leq 8$. (For any given $(r,s)$, we find finitely many central charges where more solutions exist, which depend on the choice of $\mathcal{D}$.) Therefore, at generic central charge, the field $W_{(r,s)}^{\kappa}$ is determined up to the ambiguity (2.27), and we can measure the paramter $\kappa$ using Eq. (2.22). In order to make that parameter manisfestly invariant under rescalings of $\mathcal{L}_{(r,s)}$, we can relax the normalization condition (2.20) on $\mathcal{D}$, and rewrite (some of) Eq. (2.22) in the form

$$\frac{1}{[[\mathcal{D}, \mathcal{L}_{(r,s)}]]'(\Delta_{(r,s)})} \mathcal{L}_{(r,s)} \mathcal{D} W_{(r,s)}^{\kappa} = \kappa\left(L_0 - \Delta_{(r,-s)}\right) W_{(r,s)}^{\kappa}. \tag{2.29}$$

**Degenerate fields and special values of $\kappa$**

We will now argue that two special values of $\kappa$ are singled out by degenerate fields. By a degenerate field we mean a diagonal primary field with not only a degenerate dimension of the type (2.10), but also such that the left and right null vectors vanish. We will use the notation $V_{\langle r_0, s_0 \rangle}$ for the degenerate field of conformal dimension $\Delta_{(r_0, s_0)}$.

In order to derive constraints on logarithmic representations from the existence of degenerate fields, we will study operator product expansions, a technique that dates back to the very origins of logarithmic CFT [14]. Pushing this technique to greater generality, we will apply it to degenerate fields with arbitrary indices $r_0, s_0$, and to logarithmic representations with arbitrary indices $r, s$.

Operator product expansions involving degenerate fields are constrained by fusion rules, which are simpler when written in terms of the momentum (2.9) rather than the conformal dimension:

$$V_{\langle r_0, s_0 \rangle} V_P \sim \sum_{i=-\frac{r_0-1}{2}}^{\frac{r_0-1}{2}} \sum_{j=-\frac{s_0-1}{2}}^{\frac{s_0-1}{2}} V_{P+i\beta+j\beta^{-1}}, \tag{2.30}$$

where the sums run by increments of 1. Given $r, s \in \mathbb{N}^*$, we now consider a degenerate field of the type $V_{\langle 1, s_0 \rangle}$ and a diagonal primary field $V_{P_{(r,0)}}$ such that their OPE includes the two primary field $V_{(r,\pm s)}$. (This happens if $s_0 \in s + 1 + 2\mathbb{N}$.) We consider the OPE

$$V_{\langle 1, s_0 \rangle} V_{P_{(r,0)}+\epsilon} = C_1(\epsilon)\Big[1 + f(\epsilon)\mathcal{L}_{(r,s)} + f(\epsilon)\bar{\mathcal{L}}_{(r,s)} + f(\epsilon)^2 \mathcal{L}_{(r,s)}\bar{\mathcal{L}}_{(r,s)}\Big] V_{P_{(r,s)}+\epsilon}$$
$$+ C_2(\epsilon) V_{P_{(r,-s)}+\epsilon} + \cdots. \tag{2.31}$$

Here $C_i(\epsilon)$ and $f(\epsilon)$ are OPE coefficients, which also depend on the fields' positions. We omit contributions of other primary fields, and of all descendant fields except for the three $\mathcal{L}_{(r,s)}, \bar{\mathcal{L}}_{(r,s)}$ and $\mathcal{L}_{(r,s)}\bar{\mathcal{L}}_{(r,s)}$-descendants, which become null vectors as $\epsilon \to 0$. Therefore, the coefficient $f(\epsilon)$ has a simple pole at $\epsilon = 0$.

The behaviour of our OPE will now follow from two facts:

- *The OPE is finite.* This is a consequence of the associativity of the double OPE $V_{\langle 1,s_0\rangle}V_{P_{(r,0)}+\epsilon}V_{P'}$ for a generic $P'$.

- $C_1(\epsilon)$ *has a simple zero at* $\epsilon = 0$. The coefficients $C_1(\epsilon), C_2(\epsilon)$ are determined by crossing symmetry of $\left\langle V_{P_{(r,0)}+\epsilon}V_{\langle 1,s_0\rangle}V_{P_{(r,0)}+\epsilon}V_{\langle 1,s_0\rangle}\right\rangle$ via standard analytic bootstrap methods [13], and do not depend on the particular CFT we are considering. One way to determine their behaviour is to read it from their known expressions in Liouville theory, see Section 3.3.

We deduce that the first term of the OPE (2.31) has a simple pole, which must cancel with a simple pole of the second term. This implies that $C_2(\epsilon)$ has a simple pole, such that

$$\lim_{\epsilon\to 0}\frac{C_1(\epsilon)f(\epsilon)^2}{C_2(\epsilon)} = -1\ . \tag{2.32}$$

It follows that the leading behaviour of the terms (2.31) includes a contribution from the derivative field

$$W^-_{(r,s)} = \partial_P V_{P_{(r,-s)}} - \mathcal{L}_{(r,s)}\bar{\mathcal{L}}_{(r,s)}\partial_P V_{P_{(r,s)}}\ . \tag{2.33}$$

This is a special case of the derivative field $W^\kappa_{(r,s)}$ (2.14), whose value of $\kappa$ is found by translating $P$-derivatives into $\Delta$-derivatives,

$$\boxed{\kappa^-_{(r,s)} = \frac{P_{(r,s)}}{P_{(r,s)} - P_{(r,-s)}} = \frac{s - r\beta^2}{2s}}\ . \tag{2.34}$$

Had we started with degenerate fields of the type $V_{\langle r_0,1\rangle}$ instead of $V_{\langle 1,s_0\rangle}$, we would have found derivative fields of the type

$$W^+_{(r,s)} = \partial_P V_{P_{(-r,s)}} - \mathcal{L}_{(r,s)}\bar{\mathcal{L}}_{(r,s)}\partial_P V_{P_{(r,s)}}\ , \tag{2.35}$$

whose parameter $\kappa$ is

$$\boxed{\kappa^+_{(r,s)} = \frac{P_{(r,s)}}{P_{(r,s)} - P_{(-r,s)}} = \frac{r - s\beta^{-2}}{2r}}\ . \tag{2.36}$$

In theories with both types of degenerate fields, both types of derivative fields can exist.

## 2.3 Second derivatives of null fields

We will now consider second derivative fields. Like our first derivative fields $W^\kappa_{(r,s)}$, second derivative fields appear in OPEs of degenerate fields. Higher derivative fields only appear in subleading terms of these OPEs, and we will refrain from considering them.

### Combinations of second derivatives

We introduce the combination

$$\boxed{\widetilde{W}^\kappa_{(r,s)} = \frac{1-\kappa}{2}V''_{(r,-s)} + \frac{\kappa}{2}\mathcal{L}\bar{\mathcal{L}}V''_{(r,s)}}\ , \tag{2.37}$$

and we call $\widetilde{\mathcal{W}}^\kappa_{(r,s)}$ the corresponding representation of the product of two Virasoro algebras. The field $\widetilde{W}^\kappa_{(r,s)}$ obeys

$$\left(L_0 - \Delta_{(r,-s)}\right)\widetilde{W}^\kappa_{(r,s)} = \left(\bar{L}_0 - \Delta_{(r,-s)}\right)\widetilde{W}^\kappa_{(r,s)} = W^\kappa_{(r,s)}\ , \tag{2.38}$$

$$\left(L_0 - \Delta_{(r,-s)}\right)^2\widetilde{W}^\kappa_{(r,s)} = V_{(r,-s)}\ . \tag{2.39}$$

Using an annihilation operator $\mathcal{D}$ normalized as in Eq. (2.20), and taking the second derivative of Eq. (2.16), we obtain

$$\mathcal{D}\widetilde{W}^{\kappa}_{(r,s)} = \kappa\bar{\mathcal{L}}V'_{(r,s)} \quad , \quad \bar{\mathcal{D}}\widetilde{W}^{\kappa}_{(r,s)} = \kappa\mathcal{L}V'_{(r,s)} \quad , \quad \mathcal{D}\bar{\mathcal{D}}\widetilde{W}^{\kappa}_{(r,s)} = \kappa V_{(r,s)} . \tag{2.40}$$

This leads to a closed equation for the field $\widetilde{W}^{\kappa}_{(r,s)}$,

$$\boxed{\mathcal{L}_{(r,s)}\bar{\mathcal{L}}_{(r,s)}\mathcal{D}\bar{\mathcal{D}}\widetilde{W}^{\kappa}_{(r,s)} = \kappa\left(L_0 - \Delta_{(r,-s)}\right)^2 \widetilde{W}^{\kappa}_{(r,s)}} . \tag{2.41}$$

This also leads to

$$\mathcal{L}_{(r,s)}\mathcal{D}\widetilde{W}^{\kappa}_{(r,s)} = \bar{\mathcal{L}}_{(r,s)}\bar{\mathcal{D}}\widetilde{W}^{\kappa}_{(r,s)} = \kappa W^{1}_{(r,s)} . \tag{2.42}$$

Therefore, the representation $\widetilde{\mathcal{W}}^{\kappa}_{(r,s)}$ contains both fields $W^{1}_{(r,s)}$ and $W^{\kappa}_{(r,s)}$. By linearly combining these fields, we could obtain $W^{\kappa'}_{(r,s)}$ for any value of $\kappa'$. We collectively denote these fields as $W^{*}_{(r,s)}$ in the following plot, which uses the same conventions as the plot (2.24):

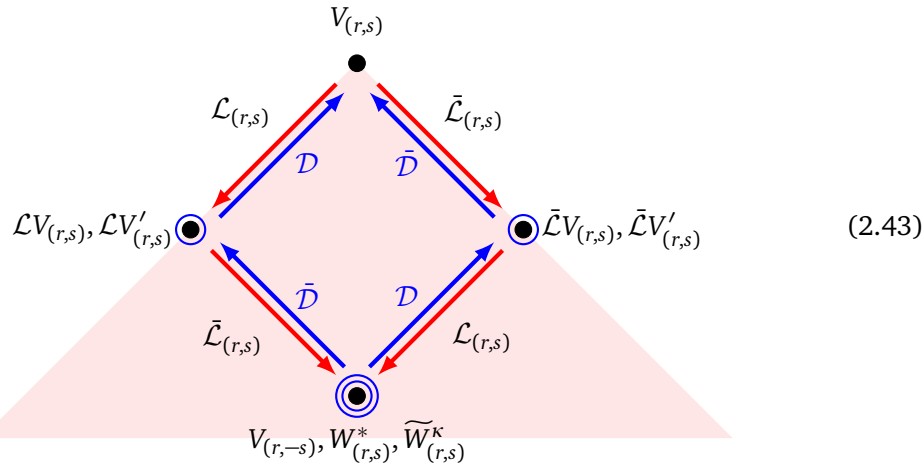

$$\tag{2.43}$$

In terms of subrepresentations, we have

$$\mathcal{V}_{(r,s)} \otimes \bar{\mathcal{V}}_{(r,s)} \subset \widetilde{\mathcal{W}}^{\kappa}_{(r,s)} \quad , \quad \mathcal{W}^{*}_{(r,s)} \subset \widetilde{\mathcal{W}}^{\kappa}_{(r,s)} . \tag{2.44}$$

Under a change of normalization that respects Eq. (2.26), the second derivative field changes as

$$\widetilde{W}^{\kappa}_{(r,s)} \to \lambda(\Delta_{(r,s)})\widetilde{W}^{\kappa}_{(r,s)} + (1-\kappa)\lambda'(\Delta_{(r,-s)})V'_{(r,-s)} + \kappa\lambda'(\Delta_{(r,s)})\mathcal{L}\bar{\mathcal{L}}V'_{(r,s)}$$
$$+ \frac{1}{2}\left[(1-\kappa)\lambda''(\Delta_{(r,-s)}) + \kappa\lambda''(\Delta_{(r,s)})\right]V_{(r,-s)} . \tag{2.45}$$

In particular, there appear fields $W^{\kappa'}_{(r,s)}$ with parameters $\kappa'$ that depend on the function $\lambda(\Delta)$. Since all these fields belong to the representation $\widetilde{\mathcal{W}}^{\kappa}_{(r,s)}$, changes of normalization amount to changes of bases.

### Degenerate fields and special values of $\kappa$

Given $r,s \in \mathbb{N}^*$, let us consider a momentum $P_0$ such that the degenerate fusion rule for the OPE $V_{\langle r_0,s_0\rangle}V_{P_0}$ (2.30) allows the four field $V_{P_{(\pm r,\pm s)}}$. For simplicity, we choose $P_0 = P_{(0,0)} = 0$

and assume $r_0 \in r + 1 + 2\mathbb{N}, s_0 \in s + 1 + 2\mathbb{N}$, but the results do not depend on this choice. We focus on the behaviour of a few terms in the OPE $V_{\langle r_0, s_0 \rangle} V_\epsilon$ as $\epsilon \to 0$,

$$V_{\langle r_0, s_0 \rangle} V_\epsilon = \sum_{\pm} \left\{ C_1(\epsilon) \Big[ 1 + f(\epsilon) \mathcal{L}_{(r,s)} + f(\epsilon) \bar{\mathcal{L}}_{(r,s)} + f(\epsilon)^2 \mathcal{L}_{(r,s)} \bar{\mathcal{L}}_{(r,s)} \Big] V_{P_{(r,s)} \pm \epsilon} \right.$$
$$\left. + C_2(\epsilon) V_{P_{(r,-s)} \pm \epsilon} \right\} + \cdots , \quad (2.46)$$

where we assume that the fields are normalized such that $V_P = V_{-P}$, which implies that the coefficients $C_i(P), f(P)$ are even functions of $P$. As in the case with first derivatives, the leading terms of the OPE cancel. The proof of that cancellation still works, and Eq. (2.32) is still valid for our coefficients $C_i(P), f(P)$, with the only difference that $C_1(\epsilon)$ now has a finite limit instead of a simple zero. The cancelling leading terms are now double poles; the simple poles also cancel by parity in $\epsilon$. The limit of our OPE therefore involves the double derivative field

$$\widetilde{W}^0_{(r,s)} = \partial_P^2 V_{P_{(r,-s)}} - \mathcal{L}_{(r,s)} \bar{\mathcal{L}}_{(r,s)} \partial_P^2 V_{P_{(r,s)}} . \quad (2.47)$$

Transforming $P$-derivatives into $\Delta$-derivatives, and using the ambiguity (2.45) for getting rid of first derivative terms, we find that $\widetilde{W}^0_{(r,s)}$ is a field of the type $\widetilde{W}^\kappa_{(r,s)}$ (2.37), for the special value of the parameter

$$\boxed{\kappa^0_{(r,s)} = \frac{P^2_{(r,s)}}{P^2_{(r,s)} - P^2_{(r,-s)}} = \frac{1}{2} - \frac{r}{4s} \beta^2 - \frac{s}{4r} \beta^{-2}} . \quad (2.48)$$

# 3 Correlation functions and conformal blocks

Let us compute correlation functions of our logarithmic fields. We will pay particular attention to the cases of two-point functions and four-point conformal blocks on the sphere. Two point functions are simple observables that capture the structure of representations, including their parameters. Four-point functions are necessary and sufficient for establishing consistency of a CFT on the sphere.

For expository reasons, we will first review two-point functions and four-point blocks for derivatives of primary fields, which are rather trivial. New results start with the two-point functions (3.21) of fields in the representation $\mathcal{W}^\kappa_{(r,s)}$ for arbitrary indices $r, s$. Then we will compute logarithmic four-point blocks that do not obey any differential equations and have no Coulomb gas integral representations: this is apparently unheard of in the earlier literature.

## 3.1 Two-point functions

In contrast to higher correlation functions, two-point functions of derivative fields cannot be computed by just differentiating two-point functions of primary fields. In a CFT with continuously varying conformal dimensions, the two-point function involves a Dirac delta function $\left\langle V_{\Delta_1}(z_1) V_{\Delta_2}(z_2) \right\rangle \propto \frac{\delta(\Delta_1 - \Delta_2)}{|z_{12}|^{4\Delta_1}}$. While we can easily deduce the simple identities

$$\Delta_1 \neq \Delta_2 \quad \Longrightarrow \quad \left\langle V^{(i)}_{\Delta_1} V^{(j)}_{\Delta_2} \right\rangle = 0 , \quad (3.1)$$

there is no well-defined procedure to recover two-point functions of $V_\Delta$ and its derivatives, which would require setting $\Delta_1 = \Delta_2$.

Instead of just differentiating two-point functions, we therefore have to differentiate the Ward identities for the two-point functions, and solve the differentiated Ward identities [15].

**Derivatives of primary fields**

Let us determine the matrix $\hat{B}_\Delta$ of two-point functions of the derivatives of $V_\Delta$ up to order $n$:

$$\hat{B}_\Delta^{ij} = \left\langle \frac{1}{(n-i)!} V_\Delta^{(n-i)} \frac{1}{(n-j)!} V_\Delta^{(n-j)} \right\rangle . \tag{3.2}$$

Here and in the rest of Section 3.1, the first field is at $z_1$ and the second field at $z_2$, i.e. we use the notation

$$\langle XY \rangle = \langle X(z_1)Y(z_2) \rangle . \tag{3.3}$$

Two-point functions of arbitrary fields obey the global Ward identities

$$\left( \partial_{z_1} + \partial_{z_2} \right) \langle XY \rangle = 0 , \tag{3.4}$$

$$\left( z_1 \partial_{z_1} + z_2 \partial_{z_2} + L_0^{(X)} + L_0^{(Y)} \right) \langle XY \rangle = 0 , \tag{3.5}$$

$$\left( z_1^2 \partial_{z_1} + z_2^2 \partial_{z_2} + 2z_1 L_0^{(X)} + 2z_2 L_0^{(Y)} + L_1^{(X)} + L_1^{(Y)} \right) \langle XY \rangle = 0 , \tag{3.6}$$

where the $L_1$-terms vanish when the fields are primaries or derivatives thereof. In this case, the global Ward identities imply $(L_0^{(X)} - L_0^{(Y)}) \langle XY \rangle = 0$. In matrix form, this equation reads

$$\hat{\Delta} \hat{B}_\Delta = \hat{B}_\Delta \hat{\Delta}^T , \tag{3.7}$$

where $\hat{\Delta}$ (2.6) is the matrix form of the action of $L_0$ on the vector of derivative field $\hat{V}_\Delta$ (2.3). The general solution of this equation is

$$\hat{B}_\Delta = f(\hat{\Delta}) \hat{B}_0 = \hat{B}_0 f(\hat{\Delta}^T) , \tag{3.8}$$

where we introduced

$$\hat{B}_0 = \begin{bmatrix} b_n & b_{n-1} & \cdots & & b_0 \\ b_{n-1} & \cdot^{\cdot^\cdot} & & b_0 & 0 \\ \vdots & & \cdot^{\cdot^\cdot} & & \vdots \\ & b_0 & & & \\ b_0 & 0 & \cdots & & 0 \end{bmatrix} , \tag{3.9}$$

for some coefficients $b_0, \ldots, b_n$, and the function $f$ is undetermined. To determine it, we use the rest of the Ward identities, and find $f(\Delta) = |z_{12}|^{-4\Delta}$, plus the requirement that the coefficients $b_i$ are $z_k$-independent. For example, in the case $n = 1$, we recover the well-known result [1]

$$\begin{bmatrix} \langle V_\Delta' V_\Delta' \rangle & \langle V_\Delta' V_\Delta \rangle \\ \langle V_\Delta V_\Delta' \rangle & \langle V_\Delta V_\Delta \rangle \end{bmatrix} = \frac{1}{|z_{12}|^{4\Delta}} \begin{bmatrix} b_1 - b_0 \log|z_{12}|^4 & b_0 \\ b_0 & 0 \end{bmatrix} . \tag{3.10}$$

In the case $n = 2$, the result is

$$\begin{bmatrix} \langle \frac{1}{2} V_\Delta'' \frac{1}{2} V_\Delta'' \rangle & \langle \frac{1}{2} V_\Delta'' V_\Delta' \rangle & \langle \frac{1}{2} V_\Delta'' V_\Delta \rangle \\ \langle V_\Delta' \frac{1}{2} V_\Delta'' \rangle & \langle V_\Delta' V_\Delta' \rangle & \langle V_\Delta' V_\Delta \rangle \\ \langle V_\Delta \frac{1}{2} V_\Delta'' \rangle & \langle V_\Delta V_\Delta' \rangle & \langle V_\Delta V_\Delta \rangle \end{bmatrix}$$

$$= \frac{1}{|z_{12}|^{4\Delta}} \begin{bmatrix} b_2 - b_1 \log|z_{12}|^4 + \frac{1}{2} b_0 \left( \log|z_{12}|^4 \right)^2 & b_1 - b_0 \log|z_{12}|^4 & b_0 \\ b_1 - b_0 \log|z_{12}|^4 & b_0 & 0 \\ b_0 & 0 & 0 \end{bmatrix} . \tag{3.11}$$

Notice that the existence of derivative fields constrains the two-point functions of the original primary field to vanish. More generally, $i + j < n \implies \left\langle V_\Delta^{(i)} V_\Delta^{(j)} \right\rangle = 0$.

Recall that two-point functions of descendants of a primary field $V_\Delta$ obey the identity

$$\left\langle \mathcal{L}_1 V_\Delta \mathcal{L}_2 V_\Delta \right\rangle = (-1)^{|\mathcal{L}_1| + |\mathcal{L}_2|} \left\langle \mathcal{L}_2 V_\Delta \mathcal{L}_1 V_\Delta \right\rangle , \tag{3.12}$$

where $\mathcal{L}_1$ and $\mathcal{L}_2$ are creation operators, and $|\mathcal{L}|$ is the level of $\mathcal{L}$, in particular $|\mathcal{L}_{(r,s)}| = rs$. For derivative fields, we observe $\left\langle V_\Delta^{(i)} V_\Delta^{(j)} \right\rangle = \left\langle V_\Delta^{(j)} V_\Delta^{(i)} \right\rangle$. Therefore, the identity (3.12) still holds if $\mathcal{L}_1, \mathcal{L}_2$ are combinations of creation operators and derivatives with respect to $\Delta$.

**Null fields**

Since the field $V_{(r,s)}$ and its null descendant $\mathcal{L} V_{(r,s)}$ are two primary fields of different dimensions, their two-point function vanishes, and so do two-point functions of their descendants. In particular,

$$\left\langle V_{(r,s)} \mathcal{L} V_{(r,s)} \right\rangle = \left\langle \mathcal{L} V_{(r,s)} \mathcal{L} V_{(r,s)} \right\rangle = 0 . \tag{3.13}$$

The non-trivial and non-vanishing quantities that will appear in two-point functions of derivative fields are

$$\rho_{(r,s)} = \frac{\partial}{\partial \Delta} \frac{\left\langle \mathcal{L}_{(r,s)} V_\Delta \mathcal{L}_{(r,s)} V_\Delta \right\rangle}{z_{12}^{-rs} z_{21}^{-rs} \left\langle V_\Delta V_\Delta \right\rangle} \Bigg|_{\Delta = \Delta_{(r,s)}} = 2 \frac{\left\langle \mathcal{L} V_{(r,s)} \mathcal{L} V'_{(r,s)} \right\rangle}{z_{12}^{-rs} z_{21}^{-rs} \left\langle V_{(r,s)} V_{(r,s)} \right\rangle} , \tag{3.14}$$

$$\omega_{(r,s)} = \frac{\partial}{\partial \Delta} \frac{\left\langle V_\Delta \mathcal{L}_{(r,s)} V_\Delta \right\rangle}{z_{12}^{-rs} \left\langle V_\Delta V_\Delta \right\rangle} \Bigg|_{\Delta = \Delta_{(r,s)}} = \frac{\left\langle V_{(r,s)} \mathcal{L} V'_{(r,s)} \right\rangle}{z_{12}^{-rs} \left\langle V_{(r,s)} V_{(r,s)} \right\rangle} . \tag{3.15}$$

The second expression for $\rho_{(r,s)}$ directly follows from the first expression, and from the identity $\left\langle \mathcal{L}_{(r,s)} V'_\Delta \mathcal{L}_{(r,s)} V_\Delta \right\rangle = \left\langle \mathcal{L}_{(r,s)} V_\Delta \mathcal{L}_{(r,s)} V'_\Delta \right\rangle$ which follows from Eq. (3.12). Similarly, the second expression for $\omega_{(r,s)}$ follows from

$$\frac{\partial}{\partial \Delta} \left\langle V_\Delta \mathcal{L}_{(r,s)} V_\Delta \right\rangle \Bigg|_{\Delta = \Delta_{(r,s)}} = \left\langle V'_{(r,s)} \mathcal{L} V_{(r,s)} \right\rangle + \left\langle V_{(r,s)} \mathcal{L} V'_{(r,s)} \right\rangle , \tag{3.16}$$

where the first term vanishes due to Eq. (3.1). One may worry that we are using the field $V'_{(r,s)}$, whose existence in principle implies $\left\langle V_{(r,s)} V_{(r,s)} \right\rangle = 0$. However, the particular correlation function where we insert $V'_{(r,s)}$ is not related to $\left\langle V_{(r,s)} V_{(r,s)} \right\rangle$ by Ward identities, so there is no inconsistency.

The quantity $\rho_{(r,s)}$ is actually well-known, and it coincides with the denominator of the conformal block's residue at $\Delta = \Delta_{(r,s)}$. Assuming the normalization

$$\mathcal{L}_{(r,s)} = L_{-1}^{rs} + \cdots , \tag{3.17}$$

we have [16, 17]

$$\rho_{(r,s)} = - \frac{\prod_{i=1-r}^{r} \prod_{j=1-s}^{s} 2 P_{(i,j)}}{2 P_{(0,0)} P_{(r,s)}} . \tag{3.18}$$

The quantity $\omega_{(r,s)}$ does not seem so well-known. Computer-assisted calculations of examples with $rs \leq 20$ suggest

$$\omega_{(r,s)} = \frac{2 P_{(r+1,s+1)}}{2 P_{(1,1)}} \frac{\prod_{i=0}^{r} \prod_{j=0}^{s} 2 P_{(i,j)}}{2 P_{(0,0)} 2 P_{(r,0)} 2 P_{(0,s)} 2 P_{(r,s)}} \prod_{i=1}^{r-1} \prod_{j=1}^{s-1} 2 P_{(i,j)} . \tag{3.19}$$

In practice, it is convenient to repeatedly use the global Ward identity (3.6), which reduces $\omega_{(r,s)}$ to the purely algebraic quantity

$$\omega_{(r,s)} = \frac{[[L_1^{rs}, \mathcal{L}_{(r,s)}]]'(\Delta_{(r,s)})}{(rs)!} \ , \tag{3.20}$$

where the polynomial $[[L_1^{rs}, \mathcal{L}_{(r,s)}]](\Delta)$ was defined in Eq. (2.16). In the formulas for $\rho_{(r,s)}$ and $\omega_{(r,s)}$, the total powers of $\Delta \sim P^2$ coincide with what we would expect from the heuristic rule $\mathcal{L}_{(r,s)} \propto \Delta^{rs}$.

**Derivatives of null fields**

Let us compute two-point functions in the representation $\mathcal{W}_{(r,s)}^{\kappa}$ (2.24). We will focus on the representation-generating logarithmic field $W_{(r,s)}^{\kappa}$, and on the primary fields $V_{(r,-s)}$ and $\bar{\mathcal{L}}V_{(r,s)}$. (The primary field $\mathcal{L}V_{(r,s)}$ can be dealt with similarly.) We normalize the creation operator $\mathcal{L}_{(r,s)}$ via Eq. (3.17). Let us show that the two-point functions are of the type

$$
\begin{bmatrix}
\left\langle W_{(r,s)}^{\kappa} W_{(r,s)}^{\kappa} \right\rangle & \left\langle W_{(r,s)}^{\kappa} V_{(r,-s)} \right\rangle & \left\langle W_{(r,s)}^{\kappa} \bar{\mathcal{L}}V_{(r,s)} \right\rangle \\
\left\langle V_{(r,-s)} W_{(r,s)}^{\kappa} \right\rangle & \left\langle V_{(r,-s)} V_{(r,-s)} \right\rangle & \left\langle V_{(r,-s)} \bar{\mathcal{L}}V_{(r,s)} \right\rangle \\
\left\langle \bar{\mathcal{L}}V_{(r,s)} W_{(r,s)}^{\kappa} \right\rangle & \left\langle \bar{\mathcal{L}}V_{(r,s)} V_{(r,-s)} \right\rangle & \left\langle \bar{\mathcal{L}}V_{(r,s)} \bar{\mathcal{L}}V_{(r,s)} \right\rangle
\end{bmatrix}
$$
$$
= \frac{1}{|z_{12}|^{4\Delta_{(r,-s)}}}
\begin{bmatrix}
b_1 - \log|z_{12}|^4 & 1 & \frac{2\omega_{(r,s)}}{\rho_{(r,s)}} z_{12}^{rs} \\
1 & 0 & 0 \\
\frac{2\omega_{(r,s)}}{\rho_{(r,s)}} z_{21}^{rs} & 0 & \frac{2}{\kappa \rho_{(r,s)}} z_{12}^{rs} z_{21}^{rs}
\end{bmatrix} . \tag{3.21}
$$

We start with the Jordan block of dimension 2 whose basis is $(V_{(r,-s)}, W_{(r,s)}^{\kappa})$. Since $L_1 W_{(r,s)}^{\kappa} \neq 0$, we may fear that we cannot directly apply Eq. (3.10). However, thanks to Eq. (3.12), the $L_1$ terms actually cancel in the Ward identiy (3.6), so that (3.10) is applicable after all. This yields the top left submatrix of size two, where we have normalized $W_{(r,s)}^{\kappa}$ so that $b_0 = 1$, and $b_1$ cannot be determined as it is not invariant under changes of bases (2.27).

Starting from the definition of $\rho_{(r,s)}$ (3.14), let us insert the action of $\bar{\mathcal{L}}_{(r,s)}$. Noticing that the relations (2.12) and (3.1) imply $\left\langle V_{(r,-s)}' V_{(r,-s)} \right\rangle = 0$, we can rewrite the numerator as a two-point function of $W_{(r,s)}^{\kappa}$ (2.14),

$$\rho_{(r,s)} = \frac{2 \left\langle W_{(r,s)}^{\kappa} V_{(r,-s)} \right\rangle}{\kappa z_{12}^{-rs} z_{21}^{-rs} \left\langle \bar{\mathcal{L}}V_{(r,s)} \bar{\mathcal{L}}V_{(r,s)} \right\rangle} \ . \tag{3.22}$$

Next, we consider the definition of $\omega_{(r,s)}$ (3.15). Inserting again the action of $\bar{\mathcal{L}}_{(r,s)}$ on all involved fields, we obtain

$$\omega_{(r,s)} = \frac{\left\langle \bar{\mathcal{L}}V_{(r,s)} W_{(r,s)}^{\kappa} \right\rangle}{\kappa z_{12}^{-rs} \left\langle \bar{\mathcal{L}}V_{(r,s)} \bar{\mathcal{L}}V_{(r,s)} \right\rangle} \ , \tag{3.23}$$

which completes the proof of Eq. (3.21).

Two-point functions of fields in the representation $\widetilde{W}_{(r,s)}^{\kappa}$ could be computed along the same lines. They would involve more complicated coefficients of the type of $\omega_{(r,s)}$ and $\rho_{(r,s)}$.

**Logarithmic couplings**

Logarithmic representations are usually parametrized by a number called the logarithmic coupling [1], which has to be related to our parameter $\kappa$. The main difference is that $\kappa$ is fully normalization-independent, while the logarithmic coupling is not. As a result, $\kappa$ is much simpler. On the other hand, the logarithmic coupling has the advantage that it can be directly read off from the two-point functions. Let us use this for relating it to $\kappa$.

In the particular normalization where $\mathcal{L}_{(r,s)} = L_{-1}^{rs} + \cdots$ and $\mathcal{D} = L_{1}^{rs} + \cdots$, the logarithmic coupling should coincide with the ratio of the $\log|z_{12}|^2$ term with the bottom right coefficient of the matrix two-point function (3.21). Calling $\gamma$ the logarithmic coupling (since the usual notation $\beta$ for that coupling already has another meaning for us), we therefore find

$$\boxed{\gamma = \rho_{(r,s)}\kappa}, \tag{3.24}$$

where $\rho_{(r,s)}$ (3.18) should be considered a normalization prefactor.

Let us compare this with some known logarithmic couplings from logarithmic minimal models [6]. **In the rest of Section 3.1, for the first and only time in this article, the central charge is not generic, but rational.** We first focus on the logarithmic minimal model $LM(2,3)$, which corresponds to $\beta^2 = \frac{3}{2}$. Due to the existence of degenerate fields, the logarithmic coupling should be $\gamma_{(r,s)}^- = \rho_{(r,s)}\kappa_{(r,s)}^-$, where $\kappa_{(r,s)}^-$ is given in Eq. (2.34). We should beware of two subtleties:

- The normalization in [6] is not $\mathcal{L}_{(r,s)} = L_{-1}^{rs} + \cdots$ but $\mathcal{L}_{(r,s)} = L_{-rs} + \cdots$, so their couplings have to be multiplied with a normalization factor.

- Our labels $(r,s)$ are the indices of a non-vanishing null vector, whereas the labels in [6] are the indices of a vanishing null vector.

Modulo these subtleties, we find that the logarithmic couplings agree:

| Coupling from [6](3.9) | Normalization factor | $\gamma_{(r,s)}^-$ |
|:---:|:---:|:---:|
| $\beta_{1,4} = -\frac{1}{2}$ | $1$ | $\gamma_{(1,1)}^- = -\frac{1}{2}$ |
| $\beta_{1,5} = -\frac{5}{8}$ | $\frac{4}{9}$ | $\gamma_{(1,2)}^- = -\frac{5}{18}$ |
| $\beta_{1,7} = -\frac{35}{3}$ | $36$ | $\gamma_{(3,1)}^- = -420$ |

$$(3.25)$$

Let us also discuss the vacuum module, which includes an identity field $V_{\langle 1,1\rangle} = V_{(1,2)}$ of dimension zero whose level one null vector vanishes, but whose level two null vector $\mathcal{L}_{(1,2)}V_{\langle 1,2\rangle}$ does not. The representation $\mathcal{W}_{(1,2)}^-$, which is characterized by the coupling $-\frac{5}{8}$, obviously differs from the vacuum module, as it does not contain the identity field. Actually, the vacuum module should contain not only the identity field, but also a Jordan block of dimension 3 [18], so our module $\widetilde{\mathcal{W}}_{(1,2)}^0$ is a better candidate. The usual definition of the vacuum module's logarithmic coupling is however not based on the Jordan block of dimension 3, but on the Jordan block of dimension 2 whose generators appear as $(\mathcal{L}V_{(r,s)}, \mathcal{L}V_{(r,s)}' = W_{(r,s)}^1)$ in the diagram (2.43). The $\kappa$-parameter for this Jordan block is not the parameter $\kappa_{(r,s)}^0$ of the full module, rather it is simply $\kappa = 1$. (Notice that the identity [18](4.6) just reads in our notations $1 = \frac{1}{2}(\frac{1}{\kappa_{(r,s)}^+} + \frac{1}{\kappa_{(r,s)}^-})$.) Therefore, the usually defined logarithmic coupling is just the normalization prefactor $\rho_{(1,2)} = -\frac{20}{9}$, divided by the normalization factor $\frac{4}{9}$ from Table (3.25). The resulting coupling $-5$, initially computed in [19], is made of nothing but normalizations: the meaningful and nontrivial structural parameter is actually $\kappa_{(1,2)}^0 = -\frac{1}{288}$.

There is also a conjecture for some of the logarithmic couplings of $LM(2, p)$ [6](5.11), which corresponds to $\beta^2 = \frac{p}{2}$. These couplings are

$$\hat{\beta}_{1,p+n} = (-1)^n \frac{p}{8} \prod_{i=-n}^{n-1} (p + 2i) = \frac{p^{2n}}{4(n-1)!^2} \gamma_{(1,n)}^- \,, \tag{3.26}$$

where the prefactor of the last expression comes from a particular choice of normalization. This equality is valid for any $1 \le n < p$, i.e. whenever the formula for $\hat{\beta}_{1,p+n}$ holds. Our expression for $\gamma_{(1,n)}^-$ should also be valid for more general values of $n$.

## 3.2 Four-point conformal blocks

If we wanted to compute four-point functions of logarithmic fields, we would of course need the corresponding logarithmic conformal blocks. These would just be derivatives of non-logarithmic conformal blocks with respect to conformal dimensions. Computing such derivatives is straightforward, because conformal blocks are analytic functions of the fields' dimensions.

However, in CFTs such as the Potts model, we are not that interested in correlation functions of logarithmic fields – fields whose existence and properties we are establishing only now. More interesting observables, which have been studied for a long time, are correlation functions of non-logarithmic fields fields, for example correlation functions of spin fields, or the cluster connectivities of Section 5. Logarithmic fields can appear as *channel fields* when we decompose such correlation functions into conformal blocks.

**Derivatives of primary fields**

Starting from a three-point function of diagonal primary fields,

$$\left\langle \prod_{i=1}^{3} V_{\Delta_i}(z_i) \right\rangle = C_{\Delta_1,\Delta_2,\Delta_3} |z_{12}|^{2(\Delta_3-\Delta_1-\Delta_2)} |z_{13}|^{2(\Delta_2-\Delta_1-\Delta_3)} |z_{23}|^{2(\Delta_1-\Delta_2-\Delta_3)} \,, \tag{3.27}$$

we deduce a three-point function involving the derivative field $V'_{\Delta_1}(z_1)$,

$$\left\langle V'_{\Delta_1}(z_1) V_{\Delta_2}(z_2) V_{\Delta_3}(z_3) \right\rangle = \left( C'_{\Delta_1,\Delta_2,\Delta_3} - C_{\Delta_1,\Delta_2,\Delta_3} \log \left| \frac{z_{12} z_{13}}{z_{23}} \right|^2 \right)$$
$$\times |z_{12}|^{2(\Delta_3-\Delta_1-\Delta_2)} |z_{13}|^{2(\Delta_2-\Delta_1-\Delta_3)} |z_{23}|^{2(\Delta_1-\Delta_2-\Delta_3)} \,. \tag{3.28}$$

This shows that the coupling of the logarithmic field $V'_{\Delta_1}$ to two diagonal primary fields $V_{\Delta_2}, V_{\Delta_3}$ involves two structure constants $C'_{\Delta_1,\Delta_2,\Delta_3}$ and $C_{\Delta_1,\Delta_2,\Delta_3}$. Let us now recall the decomposition of a four-point function into conformal blocks,

$$\left\langle \prod_{i=1}^{4} V_{\Delta_i} \right\rangle = \int d\Delta \, C_{\Delta,\Delta_1,\Delta_2} C_{\Delta,\Delta_3,\Delta_4} \mathcal{F}_\Delta \bar{\mathcal{F}}_\Delta \,. \tag{3.29}$$

Here we omit the dependence on positions $z_i$, which are spectators in our reasoning. We introduce the $s$-channel conformal block $\mathcal{F}_\Delta$ for a Verma module of dimension $\Delta$, and its value $\bar{\mathcal{F}}_\Delta$ at complex conjugate positions. For positions $(z_1, z_2, z_3, z_4) = (z, 0, \infty, 1)$, the conformal block is

$$\mathcal{F}_\Delta(z) = z^{\Delta-\Delta_1-\Delta_2} \left( 1 + \frac{(\Delta + \Delta_1 - \Delta_2)(\Delta + \Delta_4 - \Delta_3)}{2\Delta} z + O(z^2) \right) \,. \tag{3.30}$$

Let us differentiate the four-point function's integrand with respect to the channel dimension:

$$\frac{\partial}{\partial \Delta}\left(C_{\Delta,\Delta_1,\Delta_2}C_{\Delta,\Delta_3,\Delta_4}\mathcal{F}_\Delta\bar{\mathcal{F}}_\Delta\right) = C_{\Delta,\Delta_1,\Delta_2}C_{\Delta,\Delta_3,\Delta_4}\left(\mathcal{F}_\Delta\bar{\mathcal{F}}'_\Delta + \mathcal{F}'_\Delta\bar{\mathcal{F}}_\Delta\right)$$
$$+ \left(C'_{\Delta,\Delta_1,\Delta_2}C_{\Delta,\Delta_3,\Delta_4} + C_{\Delta,\Delta_1,\Delta_2}C'_{\Delta,\Delta_3,\Delta_4}\right)\mathcal{F}_\Delta\bar{\mathcal{F}}_\Delta . \quad (3.31)$$

We interpret this expression as the contribution of the channel representation generated by the field $V'_\Delta$. This contribution involves two distinct non-chiral conformal blocks $\mathcal{F}_\Delta\bar{\mathcal{F}}'_\Delta + \mathcal{F}'_\Delta\bar{\mathcal{F}}_\Delta$ and $\mathcal{F}_\Delta\bar{\mathcal{F}}_\Delta$, whose coefficients are combinations of three-point structure constants. For the channel representation generated by $V^{(n)}_\Delta$, we would similarly obtain a linear combination of the conformal blocks $\mathcal{F}_\Delta\bar{\mathcal{F}}_\Delta, (\mathcal{F}_\Delta\bar{\mathcal{F}}_\Delta)', \ldots, (\mathcal{F}_\Delta\bar{\mathcal{F}}_\Delta)^{(n)}$.

We insist that taking derivatives is only a technical trick, and that the resulting expressions are also valid if the spectrum is discrete, i.e. if the decomposition into conformal blocks is a discrete sum rather than an integral. In this context, the structure constants $C'_{\Delta_1,\Delta_2,\Delta_3}$ should not be understood as the derivative of $C_{\Delta_1,\Delta_2,\Delta_3}$, but as an independent structure constant.

**Derivatives of null fields**

The conformal block $\mathcal{F}_\Delta$ has simple poles at degenerate values of the channel dimension $\Delta \in \{\Delta_{(r,s)}\}_{r,s\in\mathbb{N}^*}$. The first pole $\Delta = \Delta_{(1,1)} = 0$ can be seen in Eq. (3.30). The behaviour near a simple pole is

$$\mathcal{F}_{\Delta_{(r,s)}+\epsilon} = \frac{R_{r,s}}{\epsilon}\mathcal{F}_{\Delta_{(r,-s)}} + \mathcal{F}^{\text{reg}}_{\Delta_{(r,s)}} + \epsilon\mathcal{F}^{(1)}_{\Delta_{(r,s)}} + O\left(\epsilon^2\right) , \quad (3.32)$$

where $R_{r,s}$ is a known, $z$-independent constant. This is the basis for Al. Zamolodchikov's recursive representation of conformal blocks. This will now allow us to determine the conformal blocks that correspond to the logarithmic representation $\mathcal{W}^\kappa_{(r,s)}$.

Let us consider two terms that may appear in the OPE

$$V_{\Delta_1}V_{\Delta_2} = c_{\Delta_{(r,s)}+\epsilon,\Delta_1,\Delta_2}\mathcal{L}_{(r,s)}\bar{\mathcal{L}}_{(r,s)}V_{\Delta_{(r,s)}+\epsilon} + c_{\Delta_{(r,-s)}+\epsilon,\Delta_1,\Delta_2}V_{\Delta_{(r,-s)}+\epsilon} + \cdots , \quad (3.33)$$

and the two corresponding terms in the four-point function's $s$-channel decomposition,

$$\left\langle\prod_{i=1}^4 V_{\Delta_i}\right\rangle = C_{\Delta_{(r,s)}+\epsilon,\Delta_1,\Delta_2}C_{\Delta_{(r,s)}+\epsilon,\Delta_3,\Delta_4}\mathcal{F}_{\Delta_{(r,s)}+\epsilon}\bar{\mathcal{F}}_{\Delta_{(r,s)}+\epsilon}$$
$$+ C_{\Delta_{(r,-s)}+\epsilon,\Delta_1,\Delta_2}C_{\Delta_{(r,-s)}+\epsilon,\Delta_3,\Delta_4}\mathcal{F}_{\Delta_{(r,-s)}+\epsilon}\bar{\mathcal{F}}_{\Delta_{(r,-s)}+\epsilon} + \cdots . \quad (3.34)$$

Due to the relation (2.12) between $V_{(r,s)}$ and $V_{(r,-s)}$, we may expect that the two terms match in the $\epsilon \to 0$ limit, whether in the OPE or in the four-point function. However, we will only assume that they match up to a factor $\chi$,

$$\lim_{\epsilon\to 0}\frac{c_{\Delta_{(r,-s)}+\epsilon,\Delta_1,\Delta_2}}{c_{\Delta_{(r,s)}+\epsilon,\Delta_1,\Delta_2}} = \lim_{\epsilon\to 0}\frac{C_{\Delta_{(r,-s)}+\epsilon,\Delta_1,\Delta_2}C_{\Delta_{(r,-s)}+\epsilon,\Delta_3,\Delta_4}}{C_{\Delta_{(r,s)}+\epsilon,\Delta_1,\Delta_2}C_{\Delta_{(r,s)}+\epsilon,\Delta_3,\Delta_4}}\frac{\epsilon^2}{R_{r,s}^2} = \chi . \quad (3.35)$$

As we saw in Section 2.2 in the case of OPEs, it can indeed happen that the two terms cancel instead of matching, so we should allow for $\chi = -1$ in addition to $\chi = 1$. What happens is that the two-point functions $\langle V_{(r,-s)}V_{(r,-s)}\rangle$ and $\langle \mathcal{L}\bar{\mathcal{L}}V_{(r,s)}\mathcal{L}\bar{\mathcal{L}}V_{(r,s)}\rangle$ can actually vanish, so the matching of the corresponding two fields determines the $\epsilon \to 0$ limits (3.35) only up to a factor $\chi$, which should not depend on $\Delta_1,\Delta_2,\Delta_3,\Delta_4$. This factor should however be the same for four-point structure constants as for OPE coefficients, because three-point functions can be computed by inserting OPEs into four-point functions.

Let us now introduce the formal linear combination of our two OPE terms,

$$
\kappa \chi c_{\Delta_{(r,s)}+\epsilon,\Delta_1,\Delta_2} \mathcal{L}_{(r,s)} \bar{\mathcal{L}}_{(r,s)} V_{\Delta_{(r,s)}+\epsilon} + (1-\kappa) c_{\Delta_{(r,-s)}+\epsilon,\Delta_1,\Delta_2} V_{\Delta_{(r,-s)}+\epsilon}
$$
$$
= c_{\Delta_{(r,-s)},\Delta_1,\Delta_2} \left( V_{(r,-s)} + \epsilon W^\kappa_{(r,s)} + \epsilon^2 \widetilde{W}^\kappa_{(r,s)} \right)
$$
$$
+ \left( \kappa \chi c'_{\Delta_{(r,s)},\Delta_1,\Delta_2} + (1-\kappa) c'_{\Delta_{(r,-s)},\Delta_1,\Delta_2} \right) \epsilon V_{(r,-s)} + O(\epsilon^2) . \quad (3.36)
$$

We have chosen a combination such that the logarithmic fields $W^\kappa_{(r,s)}$ and $\widetilde{W}^\kappa_{(r,s)}$ appear at the orders $O(\epsilon)$ and $O(\epsilon^2)$ respectively. To compute the corresponding conformal blocks, we should therefore expand the corresponding formal linear combination of the four-point function's integrand (3.29),

$$
\mathcal{Z}^\kappa_\epsilon = \kappa \chi C_{\Delta_{(r,s)}+\epsilon,\Delta_1,\Delta_2} C_{\Delta_{(r,s)}+\epsilon,\Delta_3,\Delta_4} \mathcal{F}_{\Delta_{(r,s)}+\epsilon} \bar{\mathcal{F}}_{\Delta_{(r,s)}+\epsilon}
$$
$$
+ (1-\kappa) C_{\Delta_{(r,-s)}+\epsilon,\Delta_1,\Delta_2} C_{\Delta_{(r,-s)}+\epsilon,\Delta_3,\Delta_4} \mathcal{F}_{\Delta_{(r,-s)}+\epsilon} \bar{\mathcal{F}}_{\Delta_{(r,-s)}+\epsilon} . \quad (3.37)
$$

We then have the Taylor expansion

$$
\mathcal{Z}^\kappa_\epsilon = C_{\Delta_{(r,-s)},\Delta_1,\Delta_2} C_{\Delta_{(r,-s)},\Delta_3,\Delta_4} \mathcal{F}_{\Delta_{(r,-s)}} \bar{\mathcal{F}}_{\Delta_{(r,-s)}} + \left\{ C_{\Delta_{(r,-s)},\Delta_1,\Delta_2} C_{\Delta_{(r,-s)},\Delta_3,\Delta_4} \mathcal{G}^\kappa_{(r,s)} \right.
$$
$$
\left. + \left( C_{\Delta_{(r,-s)},\Delta_1,\Delta_2} C'_{(r,s),\Delta_3,\Delta_4} + C'_{(r,s),\Delta_1,\Delta_2} C_{\Delta_{(r,-s)},\Delta_3,\Delta_4} \right) \mathcal{F}_{\Delta_{(r,-s)}} \bar{\mathcal{F}}_{\Delta_{(r,-s)}} \right\} \epsilon + O\left(\epsilon^2\right) , \quad (3.38)
$$

where $C'_{(r,s),\Delta_1,\Delta_2}$ is combination of derivatives of three-point structure constants, and we introduced the non-chiral logarithmic conformal block

$$
\mathcal{G}^\kappa_{(r,s)} = \frac{\kappa}{R_{r,s}} \left[ \mathcal{F}_{\Delta_{(r,-s)}} \bar{\mathcal{F}}^{\mathrm{reg}}_{\Delta_{(r,s)}} + \mathcal{F}^{\mathrm{reg}}_{\Delta_{(r,s)}} \bar{\mathcal{F}}_{\Delta_{(r,-s)}} \right] + (1-\kappa) \left( \mathcal{F}_{\Delta_{(r,-s)}} \bar{\mathcal{F}}_{\Delta_{(r,-s)}} \right)' . \quad (3.39)
$$

We interpret the $O(\epsilon)$ term of $\mathcal{Z}^\kappa_\epsilon$ as the contribution of the representation $\mathcal{W}^\kappa_{(r,s)}$ to a four-point function. This contribution has the same overall structure as Eq. (3.31), with two independent structure contants $C, C'$ for each three-point coupling, and two conformal blocks $\mathcal{F}_{\Delta_{(r,-s)}} \bar{\mathcal{F}}_{\Delta_{(r,-s)}}$ and $\mathcal{G}^\kappa_{(r,s)}$. The latter conformal block is however more complicated than in Eq. (3.31), and it depends on the parameter $\kappa$.

Since the logarithms come from differentiating the $z^\Delta$ prefactor of the conformal block $\mathcal{F}_\Delta(z)$ (3.30), they cancel in the combination $\frac{\bar{\mathcal{F}}^{\mathrm{reg}}_{\Delta_{(r,s)}}}{R_{r,s}} - \bar{\mathcal{F}}'_{\Delta_{(r,-s)}}$. Therefore $\mathcal{G}^\infty_{(r,s)}$ is not logarithmic, whereas $\mathcal{G}^\kappa_{(r,s)}$ is logarithmic for any finite $\kappa$. Since the field $W^\kappa_{(r,s)}$ is logarithmic if and only if $\kappa \neq \infty$, this is a basic check of our formula for $\mathcal{G}^\kappa_{(r,s)}$.

Computing the $O(\epsilon^2)$ term in the Taylor expansion (3.38) of $\mathcal{Z}^\kappa_\epsilon$, we would obtain the conformal block for the representation $\widetilde{\mathcal{W}}^\kappa_{(r,s)}$, namely

$$
\widetilde{\mathcal{G}}^\kappa_{(r,s)} = \kappa \left\{ \frac{\mathcal{F}^{\mathrm{reg}}_{\Delta_{(r,s)}} \bar{\mathcal{F}}^{\mathrm{reg}}_{\Delta_{(r,s)}}}{R^2_{r,s}} + \frac{1}{R_{r,s}} \left( \mathcal{F}^{(1)}_{\Delta_{(r,s)}} \bar{\mathcal{F}}_{\Delta_{(r,-s)}} + \bar{\mathcal{F}}^{(1)}_{\Delta_{(r,s)}} \mathcal{F}_{\Delta_{(r,-s)}} \right) \right\}
$$
$$
+ \frac{(1-\kappa)}{2} \left( \mathcal{F}_{\Delta_{(r,s)}} \bar{\mathcal{F}}_{\Delta_{(r,-s)}} \right)'' , \quad (3.40)
$$

with the coefficient $C_{\Delta_{(r,-s)},\Delta_1,\Delta_2} C_{\Delta_{(r,-s)},\Delta_3,\Delta_4}$. The $O(\epsilon^2)$ term also includes contributions of $\mathcal{G}^{\kappa'}_{(r,s)}$ and $\mathcal{F}_{\Delta_{(r,-s)}} \bar{\mathcal{F}}_{\Delta_{(r,-s)}}$, which we interpret as coming from the representation $\widetilde{\mathcal{W}}^\kappa_{(r,s)}$ again.

**Case with degenerate fields**

Let us now introduce the combination

$$\mathcal{Z}_\epsilon^- = D(P_{(r,s)} + \epsilon)\mathcal{F}_{P_{(r,s)}+\epsilon}\bar{\mathcal{F}}_{P_{(r,s)}+\epsilon} + D(P_{(r,-s)} + \epsilon)\mathcal{F}_{P_{(r,-s)}+\epsilon}\bar{\mathcal{F}}_{P_{(r,-s)}+\epsilon} \ . \tag{3.41}$$

In contrast to the formal combination $\mathcal{Z}_\epsilon^\kappa$ (3.37), we do not introduce the parameter $\kappa$. Instead, we make the assumption that degenerate fields of the type $V_{\langle 1,s_0\rangle}$ exist. This assumption leads to relations between structure constants whose arguments differ by integer multiples of $\beta^{-1}$, such as $D(P_{(r,s)} + \epsilon)$ and $D(P_{(r,-s)} + \epsilon)$. Even though our conformal blocks come from a four-point function $\left\langle \prod_{i=1}^4 V_{\Delta_i}(z_i) \right\rangle$ that needs not involve any degenerate field, the combination $\mathcal{Z}_\epsilon^-$ is analogous to the OPE (2.31). And like in that OPE, the leading terms will cancel, thanks to the relation

$$\frac{D(P_{(r,-s)} + \epsilon)}{D(P_{(r,s)} + \epsilon)} \underset{\epsilon \to 0}{\sim} -\frac{R_{r,s}\bar{R}_{r,s}}{4P_{(r,s)}^2 \epsilon^2} \ . \tag{3.42}$$

For greater generality, we allowed our four fields to be non-diagonal, in which case the residues $R_{r,s}$ and $\bar{R}_{r,s}$ of the left- and right-moving conformal blocks may differ. The factor $4P_{(r,s)}^2$ comes from translating the $\Delta$-residues $R_{r,s}$ into $P$-residues. This relation is ultimately a consequence of the analytic bootstrap equations of [20], and it can be deduced from Eqs. (2.17) and (3.8) in [21]. In the diagonal case, this relation was already observed in [5](4.19), where it was deduced from the assumption that structure constants are given by Liouville theory expressions.

The analytic bootstrap equations determine not just the leading behaviour of the ratio of structure constants as $\epsilon \to 0$, but also its value for any finite $\epsilon$. This allows us to compute the leading non-vanishing term

$$\mathcal{Z}_\epsilon^- \underset{\epsilon \to 0}{\propto} \mathcal{G}_{(r,s)}^- = 2P_{(r,s)}\left[\mathcal{F}_{\Delta_{(r,-s)}}\frac{\bar{\mathcal{F}}_{\Delta_{(r,s)}}^{\text{reg}}}{\bar{R}_{(r,s)}} + \frac{\mathcal{F}_{\Delta_{(r,s)}}^{\text{reg}}}{R_{r,s}}\bar{\mathcal{F}}_{\Delta_{(r,-s)}}\right]$$
$$- 2P_{(r,-s)}\left(\mathcal{F}_{\Delta_{(r,-s)}}\bar{\mathcal{F}}_{\Delta_{(r,-s)}}\right)' - \ell_{(r,s)}^{(1)-}\mathcal{F}_{\Delta_{(r,-s)}}\bar{\mathcal{F}}_{\Delta_{(r,-s)}} \ , \tag{3.43}$$

where the coefficient $\ell_{(r,s)}^{(1)-}$ comes from the Taylor expansion

$$\log\left(\epsilon^2 \frac{D(P_{(r,-s)} + \epsilon)}{D(P_{(r,s)} + \epsilon)}\right) = \sum_{n=0}^\infty \ell_{(r,s)}^{(n)-}\epsilon^n \ . \tag{3.44}$$

Explicitly,

$$\beta\ell_{(r,s)}^{(1)-} = -4\sum_{j=1-s}^s \left\{\psi(-2\beta^{-1}P_{(r,j)}) + \psi(2\beta^{-1}P_{(r,-j)})\right\} - 4\pi\cot(\pi s\beta^{-2})$$
$$+ \sum_{j\overset{2}{=}1-s}^{s-1}\sum_{\pm,\pm}\left\{\psi\left(\tfrac{1}{2} - \beta^{-1}(P_{(r,j)} \pm P_1 \pm P_2)\right) + \psi\left(\tfrac{1}{2} + \beta^{-1}(P_{(r,j)} \pm \bar{P}_1 \pm \bar{P}_2)\right)\right\}$$
$$+ \sum_{j\overset{2}{=}1-s}^{s-1}\sum_{\pm,\pm}\left\{\psi\left(\tfrac{1}{2} - \beta^{-1}(P_{(r,j)} \pm P_3 \pm P_4)\right) + \psi\left(\tfrac{1}{2} + \beta^{-1}(P_{(r,j)} \pm \bar{P}_3 \pm \bar{P}_4)\right)\right\}, \tag{3.45}$$

where $\psi(x) = \frac{\Gamma'(x)}{\Gamma(x)}$ is the digamma function, regularized such that $\psi(-r) = \psi(r+1)$ for $r \in \mathbb{N}$.

Comparing the non-chiral conformal block $\mathcal{G}^-_{(r,s)}$ with $\mathcal{G}^\kappa_{(r,s)}$ (3.39), we first see that the logarithmic terms do correspond to the expected value $\kappa^-_{(r,s)}$ (2.34) of the parameter $\kappa$. In addition to fixing the value of $\kappa$, the existence of degenerate fields determines the coefficient of $\mathcal{F}_{\Delta_{(r,-s)}}\bar{\mathcal{F}}_{\Delta_{(r,-s)}}$, which used to be an independent structure constant in Eq. (3.38).

Instead of degenerate fields of the type $V_{\langle 1,s_0\rangle}$, we could assume the existence of degenerate fields of the type $V_{\langle r_0,1\rangle}$. In this case, we would find conformal blocks $\mathcal{G}^+_{(r,s)} = \theta \cdot \mathcal{G}^-_{(r,s)}$ for the representation $\mathcal{W}^+_{(r,s)}$, where we introduce the operation

$$\theta \ : \ \begin{cases} \beta \to -\beta^{-1} \, , \\ r \leftrightarrow s \, , \end{cases} \tag{3.46}$$

in particular $\theta \cdot P_{(r,s)} = P_{(r,s)}$ and $\theta \cdot P_{(r,-s)} = -P_{(r,-s)}$. Assuming the existence of both types of degenerate fields, or equivalently of general degenerate fields $V_{\langle r_0,s_0\rangle}$, we can obtain both types of conformal blocks, and also conformal blocks $\widetilde{\mathcal{G}}^0_{(r,s)}$ for the representations $\widetilde{\mathcal{W}}^0_{(r,s)}$ with Jordan blocks of dimension 3. The conformal block $\widetilde{\mathcal{G}}^0_{(r,s)}$ is the leading term of $\mathcal{Z}^0_\epsilon = \sum_\pm \mathcal{Z}^-_{\pm\epsilon}$ as $\epsilon \to 0$, and we find

$$\widetilde{\mathcal{G}}^0_{(r,s)} = \left(\mathcal{F}\bar{\mathcal{F}}\right)''_{P_{(r,-s)}} - \frac{4P^2_{(r,s)}}{R_{r,s}\bar{R}_{r,s}}\left((P-P_{(r,s)})^2\mathcal{F}\bar{\mathcal{F}}\right)''_{P_{(r,s)}}$$

$$+ \left(\ell^{(1)-}_{(r,s)} - \ell^{(1)+}_{(r,s)}\right)\left(\mathcal{F}\bar{\mathcal{F}}\right)'_{P_{(r,-s)}} + \frac{4P^2_{(r,s)}}{R_{r,s}\bar{R}_{r,s}}\left(\ell^{(1)-}_{(r,s)} + \ell^{(1)+}_{(r,s)}\right)\left((P-P_{(r,s)})^2\mathcal{F}\bar{\mathcal{F}}\right)'_{P_{(r,s)}}$$

$$+ \left(2\ell^{(2)}_{(r,s)} - \ell^{(1)+}_{(r,s)}\ell^{(1)-}_{(r,s)}\right)\left(\mathcal{F}\bar{\mathcal{F}}\right)_{P_{(r,-s)}} . \tag{3.47}$$

In this formula only, the primes are derivatives with respect to $P$, not $\Delta$. We have introduced $\ell^{(1)+}_{(r,s)} = \theta \cdot \ell^{(1)-}_{(r,s)}$ and

$$\ell^{(2)}_{(r,s)} = \ell^{(2)+}_{(r,s)} = \ell^{(2)-}_{(r,s)} = -8\left\{\sum_{j=1-s}^s \sum_{i=1-r}^r \frac{1}{(2P_{(i,j)})^2} - \frac{1}{(2P_{(0,0)})^2}\right\}$$

$$- \frac{1}{2}\sum_{j \overset{2}{=} 1-s}^{s-1} \sum_{j \overset{2}{=} 1-r}^{r-1} \sum_{\pm,\pm}\left\{\frac{1}{(P_1 \pm P_2 \pm P_{(i,j)})^2} + \frac{1}{(\bar{P}_1 \pm \bar{P}_2 \pm P_{(i,j)})^2}\right.$$

$$\left. + \frac{1}{(P_3 \pm P_4 \pm P_{(i,j)})^2} + \frac{1}{(\bar{P}_3 \pm \bar{P}_4 \pm P_{(i,j)})^2}\right\} . \tag{3.48}$$

The statement here is that the coefficient $\ell^{(n)-}_{(r,s)}$ in the expansion (3.44) is a linear combination of values of $\psi^{(n-1)}$, which however simplifies to a rational function of the momenta if $n$ is even. Moreover, we have $\ell^{(n)-}_{(r,s)} \underset{n\in 2\mathbb{N}^*}{=} \ell^{(n)+}_{(r,s)}$, where $\ell^{(n)+}_{(r,s)} = \theta \cdot \ell^{(n)-}_{(r,s)}$.

The conformal block $\widetilde{\mathcal{G}}^0_{(r,s)}$ (3.47) coincides with the conformal block $\widetilde{\mathcal{G}}^\kappa_{(r,s)}$ (3.40) at $\kappa = \kappa^0_{(r,s)}$ (2.48), plus terms of the type $\mathcal{G}^{\kappa'}_{(r,s)}$ and $\mathcal{F}_{\Delta_{(r,-s)}}\bar{\mathcal{F}}_{\Delta_{(r,-s)}}$, which are now completely fixed by constraints from degenerate fields. By definition of $\widetilde{\mathcal{G}}^0_{(r,s)}$ we must have the identities $\theta \cdot \kappa^0_{(r,s)} = \kappa^0_{(r,s)}$ and $\theta \cdot \widetilde{\mathcal{G}}^0_{(r,s)} = \widetilde{\mathcal{G}}^0_{(r,s)}$, which are indeed satisfied.

## 3.3 Limits of Liouville theory four-point functions

We have been building logarithmic fields as formal derivatives of diagonal fields. In CFTs where conformal dimensions take continuous values, these derivatives need not be formal, and can

be performed in actual correlation functions, leading to specific values for the parameter $\kappa$ for logarithmic fields. We will now consider the case of Liouville theory, a nontrivial CFT with a continuous spectrum, whose structure constants are known analytically. (See [13] for a review.) In this case, the presence of degenerate fields will determine the values of the parameter $\kappa$.

**Two flavours of Liouville theory**

The properties of Liouville theory, and the analytic expression for the structure constants, depend on whether $c \leq 1$ or $c \in \mathbb{C} - (-\infty, 1)$.

In the case $c \leq 1$, the null descendants of $V_{(r,s)} = V_{\Delta_{(r,s)}}$ do not vanish [22], and the relation (2.12) holds, provided the primary fields are properly normalized. It is still possible to introduce degenerate fields $V_{\langle r_0, s_0 \rangle}$ whose null descendants vanish, but they are not obtained as limits of the primary fields $V_\Delta$. Our statements on behaviour of OPE coefficients and structure constants in Sections 2.2 and 3.2 then hold.

In the case $c \in \mathbb{C} - (-\infty, 1)$, which we will call DOZZ–Liouville theory, the null descendants of $V_{(r,s)}$ vanish, i.e. $\mathcal{L}V_{(r,s)} = \bar{\mathcal{L}}V_{(r,s)} = 0$, so that $V_{(r,s)} = V_{\langle r,s \rangle}$ is a degenerate field. Nevertheless, a diagonal primary field of dimension $\Delta_{(r,-s)}$ is obtained as [16]

$$\text{(DOZZ–Liouville theory)} \qquad V_{(r,-s)} = \mathcal{L}\bar{\mathcal{L}}V'_{(r,s)}, \tag{3.49}$$

instead of our relation (2.12). Equivalently,

$$\text{(DOZZ–Liouville theory)} \qquad \mathcal{L}_{(r,s)}\bar{\mathcal{L}}_{(r,s)}V_\Delta \underset{\Delta \to \Delta_{(r,s)}}{\sim} (\Delta - \Delta_{(r,s)})V_{(r,-s)}. \tag{3.50}$$

A zero at $\Delta = \Delta_{(r,s)}$ is present in the DOZZ three-point structure constant. Nevertheless, the OPE $\lim_{\Delta \to \Delta_{(r,s)}} V_\Delta V_{\Delta_0}$ does not vanish, thanks to poles of the structure constant. In order to build the logarithmic field $W^\kappa_{(r,s)}$ in DOZZ-Liouville theory, we should therefore do the replacement $V'_{(r,s)} \to V''_{(r,s)}$ in Eq. (2.14). Our analyses of representations, OPEs and conformal blocks then hold.

**From Liouville theory to logarithmic conformal blocks**

This article starts with the structure of logarithmic representations, and deduces conformal blocks and correlation functions. It is however possible to reverse the logic, and start with logarithmic correlation functions before looking for the corresponding representations. The reverse logic has the advantage of starting from well-known quantities, namely correlation functions in Liouville theory. And this is how we actually found the representations' structure in the first place.

Let us describe this reverse logic in more detail. There is no need to redo any calculations: whatever the logic, the formulas are the same. We start with Liouville four-point functions that involve one degenerate field,

$$\mathcal{Z}^{\text{Liouville}}_\epsilon = \left\langle V_{\langle r_0, s_0 \rangle} V_{P_{(r_2, s_2)} + \epsilon} V_{P_3} V_{P_4} \right\rangle, \tag{3.51}$$

where $V_{\langle r_0, s_0 \rangle}$ with $r_0, s_0 \in \mathbb{N}^*$ is a degenerate field, and $r_2, s_2 \in \mathbb{Z}$. We consider the $s$-channel decomposition of this four-point function into conformal blocks. The $s$-channel momentums are the type $P_{(r,s)} + \epsilon$, where the possible values of the integers $r, s \in \mathbb{Z}$ are dictated by the degenerate fusion rules (2.30). Given a value of $(r,s)$, the situation depends on how many terms of the type $(\pm r, \pm s)$ are present:

- One-term case: the conformal block $\mathcal{F}_{P_{(r,s)}+\epsilon}\bar{\mathcal{F}}_{P_{(r,s)}+\epsilon}$ remains non-logarithmic as $\epsilon \to 0$.

- Two-term case $(r,0)$&$(-r,0)$ or $(0,s)$&$(0,-s)$: the leading behaviour of the sum of the two terms as $\epsilon \to 0$ gives rise to the logarithmic non-chiral conformal block $\left(\mathcal{F}_{P_{(r,0)}}\bar{\mathcal{F}}_{P_{(r,0)}}\right)'$ or $\left(\mathcal{F}_{P_{(0,s)}}\bar{\mathcal{F}}_{P_{(0,s)}}\right)'$. This corresponds to simple logarithmic representations, with no null fields involved.

- Two-term case $(r,s)$&$(r,-s)$ with $r,s \neq 0$: the leading behaviour of the sum of the two terms as $\epsilon \to 0$ gives rise to the logarithmic non-chiral conformal block $\mathcal{G}_{(r,s)}^{-}$.

- Two-term case $(r,s)$&$(-r,s)$ with $r,s \neq 0$: the leading behaviour of the sum of the two terms as $\epsilon \to 0$ gives rise to the logarithmic non-chiral conformal block $\mathcal{G}_{(r,s)}^{+}$.

- Four-term case $(r,s)$&$(r,-s)$&$(-r,s)$&$(-r,-s)$: the leading behaviour of the sum of the four terms as $\epsilon \to 0$ gives rise to the logarithmic non-chiral conformal block $\widetilde{\mathcal{G}}_{(r,s)}^{0}$.

All these conformal blocks involve at most second derivatives of the chiral Virasoro conformal blocks $\mathcal{F}_P$. This is why we limited our investigations to second derivative fields.

**From conformal blocks to OPEs and representations**

The reader may worry that we have only obtained very special examples of our logarithmic conformal blocks, since we started with four-point functions $\mathcal{Z}_{\epsilon}^{\text{Liouville}}$ (3.51) where two fields have discrete parameters. However, the other two fields are generic, and we obtain logarithmic contributions to their OPE $V_{P_3}V_{P_4}$. This is enough for characterizing the structure of the corresponding representations. This is also enough for reconstructing logarithmic contributions to correlation functions $\left\langle \prod_{i=1}^{4} V_{P_i} \right\rangle$ of generic diagonal fields, using the OPEs $V_{P_1}V_{P_2}$ and $V_{P_3}V_{P_4}$.

In Liouville theory, how do we understand the logarithmic contributions to $\left\langle \prod_{i=1}^{4} V_{P_i} \right\rangle$? This four-point function has an $s$-channel decomposition as a integral over momentums $P \in \mathbb{R}$, with no logarithmic terms involved. In order to get logarithmic terms, we could deform the contour until it goes through values of the type $P_{(r,s)}$ with $r,s \in \mathbb{Z}$. Such contorsions would not be natural, and the logarithmic values of $P$ would anyway have measure zero.

Therefore, Liouville theory should certainly not be considered a logarithmic CFT. Rather, we found one more class of interesting limits of Liouville theory correlation functions. Other interesting limits include minimal model correlation functions [13].

# 4 The $O(n)$ model and the $Q$-state Potts model

**Torus partition functions**

The main known source of information on the spaces of states of the $O(n)$ model and of the $Q$-state Potts model is their torus partition functions. This information has limitations:

- There can be interesting fields that do not contribute to the torus partition function. For example, in the Ising model, disorder fields [23] or the fields that describe cluster connectivities [24] do not belong to the minimal model.

- In the case of fields that do contribute, the torus partition function determines the structures and multiplicities of representations in simple cases such as minimal models [23], but not in more complicated cases.

In the $O(n)$ model and the $Q$-state Potts model, the torus partition function is a combination of characters with multiplicities that are generally not positive integers [25]. This is because for non-integer $Q$ and $n$, these models are defined in terms of non-local geometrical objects. Moreover, since the torus partition function is defined as a trace of a function of the dilation generator, it does not know about off-diagonal components of that generator. Therefore, it does not know about logarithmic structures.

However, as we saw in Section 2.2, we can obtain logarithmic fields by fusing two non-logarithmic fields, including one degenerate field. This will allow us to predict the existence of logarithmic representations, even if their structure is not directly captured by the partition function.

**Primary fields**

Let us review what we know about the primary fields that appear in the $O(n)$ model and of the $Q$-state Potts model, based on their respective partition functions. The question has been analyzed in the original article [25], and more recently in [26] for the $Q$-state Potts model and in [7] for the $O(n)$ model. We do not worry about the multiplicities of the fields: rather, we focus on whether they are degenerate.

The relations between the models' parameters and the central charge are

$$n = 2\cos(\pi\beta^{-2}) \quad , \quad Q = 4\cos^2(\pi\beta^2) \,, \tag{4.1}$$

where $\beta$ was defined in Eq. (2.9). We still write $V_{\langle r_0,s_0\rangle}$ with $r_0,s_0 \in \mathbb{N}^*$ for a diagonal degenerate primary field whose left and right momentums are $P_{(r_0,s_0)}$ (2.11). Moreover, we write $V^N_{(r,s)}$ with $r,s \in \mathbb{Q}$ for a primary field whose left and right dimensions are $(\Delta,\bar{\Delta}) = (\Delta_{(r,s)},\Delta_{(r,-s)})$. The superscript $N$ stands for non-diagonal, although the spin $rs$ of $V^N_{(r,s)}$ vanishes if $r = 0$ or $s = 0$. With these notations, the primary fields are

$$O(n) \text{ model: } \left\{V_{\langle r_0,1\rangle}\right\}_{r_0\in 2\mathbb{N}+1} \cup \left\{V^N_{(r,s)}\right\}_{\substack{s\in\frac{1}{2}\mathbb{N}^* \\ r\in\frac{1}{s}\mathbb{Z}}} \,, \tag{4.2}$$

$$Q\text{-state Potts model: } \left\{V_{\langle 1,s_0\rangle}\right\}_{s_0\in\mathbb{N}^*} \cup \left\{V^N_{(r,s)}\right\}_{\substack{r\in\mathbb{N}^* \\ s\in\frac{1}{r}\mathbb{Z}}} \cup \left\{V^N_{(0,s)}\right\}_{s\in\mathbb{N}+\frac{1}{2}} \,. \tag{4.3}$$

In both models, the set of degenerate fields is closed under fusion, and generated by one basic degenerate field: $V_{\langle 3,1\rangle}$ in the $O(n)$ model, and $V_{\langle 1,2\rangle}$ in the $Q$-state Potts model.

Unless $r,s \in \mathbb{Z}^*$, there are no null vectors among the descendants of the primary field $V^N_{(r,s)}$, and the corresponding representation must be the product of left and right Verma modules $\mathcal{V}_{\Delta_{(r,s)}} \otimes \bar{\mathcal{V}}_{\Delta_{(r,-s)}}$. If however $r,s \in \mathbb{Z}^*$, then $V^N_{(r,s)}$ has a null descendant on the left if $rs > 0$ or on the right if $rs < 0$. In this case, there can be several distinct indecomposable representations that contain $V^N_{(r,s)}$. In the case $r,s > 0$, the possibilities include

- $\mathcal{V}_{\Delta_{(r,s)}} \otimes \bar{\mathcal{V}}_{\Delta_{(r,-s)}}$,

- $\dfrac{\mathcal{V}_{\Delta_{(r,s)}}}{\mathcal{V}_{\Delta_{(r,-s)}}} \otimes \bar{\mathcal{V}}_{\Delta_{(r,-s)}}$, i.e. the product of a degenerate representation with a Verma module,

- $\mathcal{W}^\kappa_{(r,s)}$, our logarithmic representation with Jordan blocks of dimension 2.

We will now determine which representation is the right one.

**Logarithmic structures**

Let us start with the $Q$-state Potts model. We have primary fields of the type $V_{\langle 1,s_0\rangle}$ and $V^N_{(r,0)}$. These are the fields that appear in the $\epsilon \to 0$ limit of the OPE (2.31), which leads to logarithmic fields of the type $W^-_{(r,s)}$. This suggests that for any $(r,s) \in \mathbb{N}^* \times \mathbb{Z}^*$, the fields $V^N_{(r,s)}$ and $V^N_{(r,-s)}$ of the Potts model are part of the same logarithmic representation $\mathcal{W}^-_{(r,s)}$ (2.24), where they are called $\mathcal{L}V_{(r,s)}$ and $\bar{\mathcal{L}}V_{(r,s)}$.

In the $O(n)$ model, we have primary fields of the type $V_{\langle r_0,1\rangle}$ and $V^N_{(0,s)}$. We therefore expect logarithmic representations of the type $\mathcal{W}^+_{(r,s)}$. In the case $(r,s) = (1,1)$, let us compare this with the results of [7]. Our field $W^+_{(1,1)}$ is called $A$ in [7] (Section 5.2), and it obeys $\frac{1}{2}L_{-1}L_1 A = \kappa(L_0 - 1)A$ in agreement with Eq. (2.22). The parameter $\kappa$ is called $-s^2$ in [7], and it takes the value $\kappa^+_{(1,1)} = \frac{1-\beta^{-2}}{2}$ in agreement with Eq. (2.36). The so-called currents are $J = \mathcal{L}V_{(1,1)}$ and $\bar{J} = \bar{\mathcal{L}}V_{(1,1)}$. They are called currents because their conformal dimensions are $(\Delta, \bar{\Delta}) = (1,0)$ and $(0,1)$ respectively. However, their level one null vectors do not vanish, $\bar{\partial}J = \partial \bar{J} = V_{(1,-1)} \neq 0$.

To summarize, we propose the following logarithmic subspaces of the spectrums of the $Q$-state Potts model and $O(n)$ model,

$$\boxed{\bigoplus_{r,s=1}^{\infty} \mathcal{W}^-_{(r,s)} \subset \mathcal{S}^{Q\text{-state Potts model}}} \quad , \quad \boxed{\bigoplus_{r,s=1}^{\infty} \mathcal{W}^+_{(r,s)} \subset \mathcal{S}^{O(n)\text{ model}}} \,, \tag{4.4}$$

where we however do not know the multiplicities of the representations.

**Values of the parameters $n$ and $Q$**

The allowed values of the parameters $n$ and $Q$, and the corresponding values of $\beta^2$ (4.1), are traditionally given as [25]

$$-2 \leq c \leq 1 \quad , \quad \frac{1}{2} \leq \beta^2 \leq 1 \quad , \quad \begin{cases} -2 \leq n \leq 2\,, \\ 0 \leq Q \leq 4\,, \end{cases} \tag{4.5}$$

although it is known that analytic continuations are possible [26]. From the point of view of conformal field theory, the only hard limit is the convergence of the operator product expansions, which requires that conformal dimensions be bounded from below. Given the models' primary fields, this condition amounts to [27]

$$\Re c < 13 \iff \Re \beta^2 > 0 \,. \tag{4.6}$$

The allowed region is therefore vastly larger than the complex $n$-plane or the complex $Q$-plane. In order to cover these complex planes, it is enough to consider the following fundamental domains:

$$n \in \mathbb{C} \quad \iff \quad 1 \leq \Re \beta^{-2} < 2 \quad \iff \quad \Re \Delta_{(1,2)} < 1 \leq \Re \Delta_{(1,3)} \,, \tag{4.7}$$

$$Q \in \mathbb{C} \quad \iff \quad \frac{1}{2} < \Re \beta^2 \leq 1 \quad \iff \quad \Re \Delta_{(3,1)} \leq 1 < \Re \Delta_{(5,1)} \,. \tag{4.8}$$

(See [27] for a picture of the $Q$-state Potts model's fundamental domain in the complex $c$-plane.) We have rewritten the boundaries of the fundamental domains in terms of conditions for certain fields to be relevant. Curiously, the $Q$-state Potts model's fundamental domain is related to the relevance of degenerate fields that appear in the $O(n)$ model, and vice-versa.

It is therefore possible to analytically continue the models beyond the complex $n, Q$-planes. Nothing dramatic happens to the conformal field theories, but their statistical interpretations may change.

# 5 Four-point connectivities in the $Q$-state Potts model

Our determination of logarithmic structures in the $Q$-state Potts model relies on plausible but unproven assumptions on the existence and properties of degenerate fields. In order to test these assumptions and the logarithmic structures themselves, we will look for solutions of the crossing symmetry equations based on our logarithmic conformal blocks.

Four-point connectivities in the $Q$-state Potts model were recently computed using a semi-analytic conformal bootstrap approach [8]. Crossing symmetry could be checked to a good precision, which was only limited by the lack of knowledge of logarithmic conformal blocks. If our logarithmic structures are correct, they should allow us to bootstrap connectivities to a precision that is only limited by numerical artefacts.

## 5.1 Logarithmic structures in four-point connectivities

According to a longstanding conjecture [28], connectivities in the critical $Q$-state Potts model coincide with correlation functions of primary fields with the left and right dimension $\Delta_{(0,\frac{1}{2})}$. This conjecture was of little practical help for determining four-point connectivities, until it was complemented with another conjecture on the decompositions of the four-point functions into conformal blocks [29].

**Decomposing four-point connectivities into conformal blocks**

Let us schematically write a decomposition of a four-point connectivity $P(z_1, z_2, z_3, z_4)$:

$$P(z_k) = \sum_{i \in \mathcal{S}^{(c)}} D_i^{(c)} \mathcal{G}_i^{(c)}(z_k) \,, \tag{5.1}$$

where $D_i^{(c)}$ is a four-point structure constant, $\mathcal{G}_i^{(c)}(z_k)$ is a four-point conformal block in the channel $c \in \{s, t, u\}$ for four fields of dimension $\Delta_{(0,\frac{1}{2})}$, and $\mathcal{S}^{(c)}$ is the spectrum of the connectivity in that channel, i.e. a set of representations of the product of the left and right Virasoro algebras.

There are four independent connectivities $P_{aaaa}(z_k), P_{aabb}(z_k), P_{abab}(z_k), P_{abba}(z_k)$. Permutations of the positions $z_k$ leave $P_{aaaa}(z_k)$ invariant, and exchange the other connectivities. Let us call $\mathcal{S}^{\sigma}$ the $s$-channel spectrum of the connectivity $P_{\sigma}$. By permutation symmetry, we have $\mathcal{S}^{abab} = \mathcal{S}^{abba}$, and the connectivity $P_{aabb}$ has the spectrums $\mathcal{S}^{aabb}, \mathcal{S}^{abab}$ and $\mathcal{S}^{abab}$ in the $s$, $t$ and $u$ channels respectively.

The conjecture for the spectrums is based on a lattice discretization of the model [29], and suffers from the same shortcoming as the determination of the $Q$-state Potts model's spectrum based on the torus partition function: it does not predict the full structure of the representations, but only their primary fields. We therefore have the following three subsets of the model's set of primary fields (4.3):

$$\mathcal{S}^{aaaa} = \left\{ V_{(r,s)}^N \right\}_{\substack{r \in 2\mathbb{N}^* \\ s \in \frac{2}{r}\mathbb{Z}}} \cup \left\{ V_{(0,s)}^N \right\}_{s \in \mathbb{N} + \frac{1}{2}} \,, \tag{5.2}$$

$$\mathcal{S}^{aabb} = \left\{ V_{(r,s)}^N \right\}_{\substack{r \in 2\mathbb{N}^* \\ s \in \frac{2}{r}\mathbb{Z}}} \cup \left\{ V_{(0,s)}^N \right\}_{s \in \mathbb{N} + \frac{1}{2}} \cup \left\{ V_{\langle 1,s_0 \rangle} \right\}_{s_0 \in \mathbb{N}^*} \,, \tag{5.3}$$

$$\mathcal{S}^{abab}, \mathcal{S}^{abba} = \left\{ V_{(r,s)}^N \right\}_{\substack{r \in 2\mathbb{N}^* \\ s \in \frac{1}{r}\mathbb{Z}}} \,. \tag{5.4}$$

**Singularities of conformal blocks**

Recall that the $s$-channel conformal block $\mathcal{F}_\Delta$ for a Verma module of dimension $\Delta$ has simple poles at degenerate values $\Delta \in \{\Delta_{(r,s)}\}_{r,s \in \mathbb{N}^*}$. In a four-point function of fields with dimensions

$\Delta_{(0,\frac{1}{2})}$, some of these poles have vanishing residues, and are therefore actually absent. The pole at $\Delta = \Delta_{(r,s)}$ has a vanishing residue whenever the fusion rule (2.30) of the degenerate field $V_{\langle r,s \rangle}$ allows the fusion $V_{\langle r,s \rangle} V_{P_{(0,\frac{1}{2})}} \to V_{P_{(0,\frac{1}{2})}}$ or $V_{-P_{(0,\frac{1}{2})}}$, i.e. whenever $r$ is odd.

Any conformal block that appears in the decomposition of a four-point connectivity must of course be finite. We will now see that this basic criterion gives us hints on the structure of the spectrum.

The degenerate fields $V_{\langle 1,s_0 \rangle}$ that are conjectured to appear in the spectrum $\mathcal{S}^{aabb}$ have an odd first index, so that $\mathcal{F}_{\Delta_{(1,s_0)}}$ is finite, and the corresponding conformal block in the decomposition of the connectivity $P^{aabb}(z_k)$ (5.1) may well be of the type

$$\mathcal{G}^{aabb}_{\langle 1,s_0 \rangle} = \mathcal{F}_{\Delta_{(1,s_0)}} \bar{\mathcal{F}}_{\Delta_{(1,s_0)}} . \tag{5.5}$$

This is consistent with the claim that $V_{\langle 1,s_0 \rangle}$ is a diagonal degenerate field, i.e. a primary field that generates a degenerate representation $\frac{\mathcal{V}_{\Delta_{(1,s_0)}}}{\mathcal{V}_{\Delta_{(1,-s_0)}}} \otimes \frac{\bar{\mathcal{V}}_{\Delta_{(1,s_0)}}}{\bar{\mathcal{V}}_{\Delta_{(1,-s_0)}}}$. (In contrast to characters, four-point conformal blocks do not see the difference between Verma modules and their degenerate quotients.)

The conjectured spectrums $\mathcal{S}^{aaaa}$, $\mathcal{S}^{aabb}$ and $\mathcal{S}^{abab}$ also contain primary fields of the type $V^N_{(r,s)}$ with $(r,s) \in 2\mathbb{N}^* \times \mathbb{Z}^*$. Since the first index is even, the conformal block $\mathcal{F}_{\Delta_{(r,s)}}$ is infinite. This rules out the possibility that the corresponding representation could be a Verma module or a degenerate representation, and suggests that more complicated structures are required. Our claim is that the correct conformal blocks are of the type $\mathcal{G}^-_{(r,s)}$ (3.43).

## 5.2 Linear relations between four-point structure constants

The basic idea of the conformal bootstrap method is that the decomposition (5.1) of a given four-point function should not depend on the channel. The equality between the $s$, $t$ and $u$-channels is called crossing symmetry. Assuming that we know the spectrum, crossing symmetry amounts to linear equations for the four-point structure constants, which we will solve numerically. This flavour of the conformal bootstrap was introduced in [30], and may be called semi-analytic in contrast to situations where the spectrum is itself an unknown (numerical bootstrap) or where the structure constants are known analytically too (analytic bootstrap).

However, in our case, the structure constants are not quite independent unknowns. The degenerate fields that allowed us to predict the nontrivial conformal blocks, also lead to linear relations between certain structure constants [20]. These relations may be viewed as emanating from an "interchiral" symmetry algebra that is larger than the product of the left and right Virasoro algebra [8].

### The relations

Let us call $D^{\sigma}_{(r,s)}$ the four-point structure constants for the primary fields $V^N_{(r,s)}$ in the spectrums $\mathcal{S}^{\sigma}$ (5.2)-(5.4), and $D^{aabb}_{\langle 1,s_0 \rangle}$ the structure constants for the diagonal degenerate fields in $\mathcal{S}^{aabb}$.

Due to the known permutation properties of structure constants and conformal blocks [13], we have the relations

$$D^{abab}_{(r,s)} = (-1)^{rs} D^{abba}_{(r,s)} , \tag{5.6}$$

since $rs = \Delta_{(r,-s)} - \Delta_{(r,s)}$ is the conformal spin of $V^N_{(r,s)}$. Moreover, we are considering four-point functions of four spinless primary fields. This implies that the crossing symmetry equations are invariant under the exchange of the left and right quantities, and therefore the relations

$$D^{\sigma}_{(r,s)} = D^{\sigma}_{(r,-s)} . \tag{5.7}$$

From the definition of four-point connectivities, it is also possible to predict [27]

$$D^{aabb}_{(0,\frac{1}{2})} = -D^{aaaa}_{(0,\frac{1}{2})} \,. \tag{5.8}$$

Let us now move to the relations that follow from the existence of the degenerate field $V_{\langle 1,2 \rangle}$. In general four-point functions, these relations would determine the ratios $\frac{D^{\sigma}_{(r,s+2)}}{D^{\sigma}_{(r,s)}}$ [20]. However, in a four-point function of fields with the particular dimension $\Delta_{(0,\frac{1}{2})}$, we have slightly more powerful relations, which determine how structure constants behave under shifts of the second index by one unit, rather than the usual two units [8]:

$$\frac{D^{\sigma}_{(r,s+1)}}{D^{\sigma}_{(r,s)}} = 2^{(2+2\omega)r - \frac{4s+2}{\beta^2}} \frac{\Gamma(\frac{1-r}{2} + \frac{s}{2\beta^2})}{\Gamma(\frac{2-r}{2} + \frac{s}{2\beta^2})} \frac{\Gamma(\frac{\omega r}{2} - \frac{s}{2\beta^2})}{\Gamma(\frac{1+\omega r}{2} - \frac{s}{2\beta^2})} \frac{\Gamma(\frac{1-r}{2} + \frac{s+1}{2\beta^2})}{\Gamma(\frac{-r}{2} + \frac{s+1}{2\beta^2})} \frac{\Gamma(\frac{2+\omega r}{2} - \frac{s+1}{2\beta^2})}{\Gamma(\frac{1+\omega r}{2} - \frac{s+1}{2\beta^2})} \,, \tag{5.9}$$

where $\omega = -1$ for our non-diagonal fields $V^N_{(r,s)}$, and $\omega = 1$ for diagonal fields with dimensions $\Delta = \bar{\Delta} = \Delta_{(r,s)}$, such as $V_{\langle 1,s_0 \rangle}$. For $s \in \{0,-1\}$, this shift equation involves the value of the Gamma function at its poles. In the non-diagonal case however, these singularities cancel among the factors, and the ratio can be rewritten in a manifestly finite manner,

$$\frac{D^{\sigma}_{(r,s+1)}}{D^{\sigma}_{(r,s)}} \underset{s \in \{0,-1\}}{=} \left( \frac{2^{1-\frac{2}{\beta^2}}}{|r|} \frac{\Gamma(\frac{1-r}{2} + \frac{1}{2\beta^2}) \Gamma(\frac{2-r}{2} - \frac{1}{2\beta^2})}{\Gamma(\frac{-r}{2} + \frac{1}{2\beta^2}) \Gamma(\frac{1-r}{2} - \frac{1}{2\beta^2})} \right)^{2s+1} \,. \tag{5.10}$$

We insist that the shift equations also hold in the case $r,s \in \mathbb{Z}^*$ i.e. if $V^N_{(r,s)}$ belongs to a logarithmic representation. The validity of these equations indeed only depends on $V^N_{(r,s)}$ being a primary field.

Our formulas for the ratios are equivalent to the formulas in [8] (Section 4.1). Cosmetic differences come down to notations and to our use of the Gamma function duplication formula. We will now rederive these ratios by following the logic of [8], while trying to streamline the derivation.

**The derivation**

Let us denote two- and three-point structure constants as $\langle V_i V_i \rangle \sim B_i$ and $\langle V_i V_j V_k \rangle \sim C_{ijk}$. Then an OPE reads $V_i V_j \sim \sum_k \frac{C_{ijk}}{B_k} V_k$, and four-point structure constants that appear in $s$ and $t$-channel decompositions of four-point functions read

$$s\text{-channel:} \quad \phantom{x} \qquad \rightarrow d^{(s)}_{\epsilon} = \frac{C_{12\epsilon} C_{\epsilon 34}}{B_{\epsilon}} \,, \tag{5.11}$$

$$t\text{-channel:} \quad \phantom{x} \qquad \rightarrow d^{(t)}_{\eta} = \frac{C_{23\eta} C_{41\eta}}{B_{\eta}} \,. \tag{5.12}$$

We normalize the degenerate field $V_{\langle 1,1 \rangle}$ as an identity field, i.e. $C_{\langle 1,1 \rangle ii} = B_i$ and $B_{\langle 1,1 \rangle} = 1$.

We consider four-point functions that involve at least one degenerate field, such as $\left\langle V_{\langle 1,2 \rangle} \prod_{i=2}^4 V^N_{(r_i,s_i)} \right\rangle$ or $\left\langle V_{\langle 1,2 \rangle} V^N_{(r,s)} V^N_{(r,s)} V_{\langle 1,2 \rangle} \right\rangle$. In these cases, only two fields can appear in each channel, as dictated by the OPEs $V_{\langle 1,2 \rangle} V^N_{(r,s)} \sim \sum_{\epsilon = \pm} V^N_{(r,s+\epsilon)}$ and $V_{\langle 1,2 \rangle} V_{\langle 1,2 \rangle} \sim V_{\langle 1,1 \rangle} + V_{\langle 1,3 \rangle}$. Crossing symmetry and single-valuedness determine the ratios of all involved four-point structure constants, and in particular [20]

$$\frac{d^{(t)}_{\eta}}{d^{(s)}_{\epsilon}} = -\epsilon \omega \eta \frac{F_{\epsilon,\eta}}{\bar{F}_{\epsilon,-\omega\eta}} \det \bar{F} \qquad \text{with} \qquad \det \bar{F} = -\frac{\bar{P}_2}{\bar{P}_4} \,. \tag{5.13}$$

Here the $s$- and $t$-channel fields are labelled by discrete indices $\epsilon, \eta = \pm$. We define $\omega = +$ if the fourth field is $V_{\langle 1,2 \rangle}$, and $\omega = -$ if the fourth field is $V_{(r_4,s_4)}^N$. The fusing matrix elements are

$$F_{\epsilon,\eta} = \frac{\Gamma(1 - 2\beta^{-1}\epsilon P_2)\Gamma(2\beta^{-1}\eta P_4)}{\prod_\pm \Gamma(\frac{1}{2} - \beta^{-1}\epsilon P_2 \pm \beta^{-1}P_3 + \beta^{-1}\eta P_4)} \ . \tag{5.14}$$

We first consider the four-point function $\left\langle V_{\langle 1,2 \rangle} V_{(r,s)}^N V_{(0,\frac{1}{2})}^N V_{(0,\frac{1}{2})}^N \right\rangle$, and focus on the following $s$- and $t$-channel terms:

$$\tag{5.15}$$

In this case, Eq. (5.13) gives us $\frac{d_-^{(t)}}{d_+^{(s)}} = -\frac{\bar{P}}{P_{(0,\frac{1}{2})}} \frac{F_{+-}}{\bar{F}_{+-}}$, explicitly

$$\frac{C_{\langle 1,2 \rangle (0,\frac{1}{2})(0,\frac{1}{2})} C_{(r,s)(0,\frac{1}{2})(0,\frac{1}{2})}}{C_{\langle 1,2 \rangle (r,s)(r,s+1)} C_{(r,s+1)(0,\frac{1}{2})(0,\frac{1}{2})}} \frac{B_{(r,s+1)}}{B_{(0,\frac{1}{2})}}$$
$$= 2^{1-2\beta^{-1}(P-\bar{P})}\gamma(\tfrac{1}{2\beta^2}) \frac{\Gamma(1 - \beta^{-1}P)}{\Gamma(-\beta^{-1}\bar{P})} \frac{\Gamma(\frac{1}{2} - \beta^{-1}\bar{P} - \frac{1}{2\beta^2})}{\Gamma(\frac{1}{2} - \beta^{-1}P + \frac{1}{2\beta^2})} \ , \tag{5.16}$$

where we denote the left and right momentums of $V_{(r,s)}^N$ as $(P, \bar{P}) = (P_{(r,s)}, P_{(r,-s)})$, and introduce $\gamma(x) = \frac{\Gamma(x)}{\Gamma(1-x)}$. We next consider the four-point function $\left\langle V_{\langle 1,2 \rangle} V_{(r,s)}^N V_{(r,s)}^N V_{\langle 1,2 \rangle} \right\rangle$, and focus on the following $s$- and $t$-channel terms:

$$\tag{5.17}$$

In this case, Eq. (5.13) gives us $\frac{d_-^{(t)}}{d_+^{(s)}} = -\frac{\bar{P}}{P_{\langle 1,2 \rangle}} \frac{F_{+-}}{\bar{F}_{++}}$, explicitly

$$\frac{B_{\langle 1,2 \rangle} B_{(r,s)} B_{(r,s+1)}}{C_{\langle 1,2 \rangle (r,s)(r,s+1)}^2} = 2^{4\beta^{-2}-3}\gamma(\beta^{-2} - \tfrac{1}{2}) \frac{\Gamma(1 - 2\beta^{-1}P)}{\Gamma(-2\beta^{-1}P + \beta^{-2})} \frac{\Gamma(1 - 2\beta^{-1}\bar{P} - \beta^{-2})}{\Gamma(-2\beta^{-1}\bar{P})} \ . \tag{5.18}$$

We finally consider the four-point function $\left\langle V_{\langle 1,2 \rangle} V_{(0,\frac{1}{2})}^N V_{(0,\frac{1}{2})}^N V_{\langle 1,2 \rangle} \right\rangle$, and focus on the following $s$- and $t$-channel terms:

$$\tag{5.19}$$

In this case, Eq. (5.13) gives us $\frac{d_{-}^{(t)}}{d_{-}^{(s)}} = -\frac{P_{(0,\frac{1}{2})}}{P_{\langle 1,2\rangle}}\frac{F_{--}}{\bar{F}_{-+}}$, explicitly

$$\frac{B_{(0,\frac{1}{2})}^2 B_{\langle 1,2\rangle}}{C_{\langle 1,2\rangle(0,\frac{1}{2})(0,\frac{1}{2})}^2} = 2^{4\beta^{-2}-3}\gamma(\beta^{-2}-\tfrac{1}{2})\gamma(1-\tfrac{1}{2\beta^2})^2 \; . \tag{5.20}$$

We are interested in the four-point structure constant $D_{(r,s)} = \frac{C_{(r,s)(0,\frac{1}{2})(0,\frac{1}{2})}^2}{B_{(r,s)}}$. Combining the square of Eq. (5.16) with Eq. (5.18) and Eq. (5.20), we obtain the expression for $\frac{D_{(r,s+1)}}{D_{(r,s)}}$ that was written in Eq. (5.9) with $\omega = -1$. For the case $\omega = 1$, it suffices to set $(P,\bar{P}) = (P_{(r,s)},-P_{(r,s)})$ instead of $(P_{(r,s)},P_{(r,-s)})$.

**Interchiral conformal blocks and reduced spectrums**

Using the linear relations (5.7) and (5.9) between four-point structure constants, we can rewrite $s$-channel decompositions of four-point connectivities (5.1) such that the only unknown coefficients are $D_{(r,s)}^{\sigma}$ with $0 \leq s \leq \frac{1}{2}$. In this rewriting, $D_{(r,s)}^{\sigma}$ is the coefficient of an infinite linear combinations of conformal blocks, which was called an interchiral conformal block in [8]:

$$\mathcal{H}_{(r,s)} \underset{0<s\leq\frac{1}{2}}{=} \sum_{s'\in(s+\mathbb{Z})\cup(-s+\mathbb{Z})} \frac{D_{(r,s')}^{\sigma}}{D_{(r,s)}^{\sigma}}\mathcal{F}_{\Delta_{(r,s')}}\bar{\mathcal{F}}_{\Delta_{(r,-s')}} \; , \tag{5.21}$$

$$\mathcal{H}_{(r,0)} = \mathcal{F}_{\Delta_{(r,0)}}\bar{\mathcal{F}}_{\Delta_{(r,0)}} + \sum_{s'=1}^{\infty} \frac{D_{(r,s')}^{\sigma}}{D_{(r,0)}^{\sigma}}\mathcal{G}_{(r,s')}^{-} \; , \tag{5.22}$$

$$\mathcal{H}_{\langle 1,1\rangle} = \sum_{s'=1}^{\infty} \frac{D_{\langle 1,s'\rangle}^{\sigma}}{D_{\langle 1,1\rangle}^{\sigma}}\mathcal{F}_{\Delta_{(1,s')}}\bar{\mathcal{F}}_{\Delta_{(1,s')}} \; , \tag{5.23}$$

where $\mathcal{F}_{\Delta}$ is a standard Virasoro conformal block, and $\mathcal{G}_{(r,s)}^{-}$ (3.43) is a logarithmic conformal block. Neither the ratios of structure constants, nor therefore the interchiral conformal blocks, depend on $\sigma$. The spectrums (5.2)-(5.4) can then be reduced to the fields whose second indices obey $0 \leq s \leq \frac{1}{2}$,

$$\mathcal{S}_{\text{reduced}}^{aaaa} = \left\{V_{(r,s)}^N\right\}_{\substack{r\in 2\mathbb{N}^* \\ s\in\frac{2}{r}\mathbb{N}\cap[0,\frac{1}{2}]}} \cup \left\{V_{(0,\frac{1}{2})}^N\right\} \; , \tag{5.24}$$

$$\mathcal{S}_{\text{reduced}}^{aabb} = \left\{V_{(r,s)}^N\right\}_{\substack{r\in 2\mathbb{N}^* \\ s\in\frac{2}{r}\mathbb{N}\cap[0,\frac{1}{2}]}} \cup \left\{V_{(0,\frac{1}{2})}^N\right\} \cup \left\{V_{\langle 1,1\rangle}\right\} \; , \tag{5.25}$$

$$\mathcal{S}_{\text{reduced}}^{abab}, \mathcal{S}_{\text{reduced}}^{abba} = \left\{V_{(r,s)}^N\right\}_{\substack{r\in 2\mathbb{N}^* \\ s\in\frac{1}{r}\mathbb{N}\cap[0,\frac{1}{2}]}} \; , \tag{5.26}$$

and we write the four-point connectivities as

$$P^{\sigma} = \sum_{V\in\mathcal{S}_{\text{reduced}}^{\sigma}} D_V^{\sigma}\mathcal{H}_V \; , \tag{5.27}$$

where we abuse notations by identifying a primary field $V$ with its indices.

## 5.3 Semi-analytic bootstrap

Let us determine the four-point structure constants $D_i^{\sigma}$ by solving crossing symmetry equations for the connectivities $P_{\sigma}$ with $\sigma \in \{aaaa, aabb, abab, abba\}$. Given a connectivity and two

different channels $c, c' \in \{s, t, u\}$, we have the crossing symmetry equation

$$\sum_{i \in \mathcal{S}^{(c)}_{\text{reduced}}} D_i^{(c)} \mathcal{H}_i^{(c)}(z_k) = \sum_{i' \in \mathcal{S}^{(c')}_{\text{reduced}}} D_{i'}^{(c')} \mathcal{H}_{i'}^{(c')}(z_k) \,, \tag{5.28}$$

for any values of the positions $z_k$. We are not so much interested in the solutions themselves as in their existence, which tests several things at once:

- the identification of connectivities with correlation functions,

- the existence of degenerate fields,

- Jacobsen and Saleur's spectrums $\mathcal{S}^{aaaa}, \mathcal{S}^{aabb}$ and $\mathcal{S}^{abab}$,

- our conformal blocks $\mathcal{G}^-_{(r,s)}$ and the structure of logarithmic representations.

**Numerical implementation**

Using Zamolodchikov's recursive representation, the interchiral conformal blocks have a series expansion of the type

$$\mathcal{H}_i^{(c)}(z_k) = \mathcal{H}_0^{(c)}(z_k) \left| q^{(c)} \right|^{\Delta_i + \bar{\Delta}_i} \sum_{N=0}^{\infty} h_{i,N}\left( q^{(c)} \right) \left| q^{(c)} \right|^N \,. \tag{5.29}$$

Here $\mathcal{H}_0^{(c)}(z_k)$ is an $i$-independent prefactor, the nome $q^{(c)}$ is a function of $(z_1, z_2, z_3, z_4)$ that depends on the channel $c$, and the coefficients $h_{i,N}(q)$ is a polynomially bounded function of $\frac{q}{|q|}$ and $\log q$. In order to write the connectivities as finite sums, we introduce a cutoff $N_{\max}$ and truncate the conformal blocks' expansions to $N \leq N_{\max}$, while also truncating the sums over reduced spectrums to $\Re(\Delta_i + \bar{\Delta}_i) \leq N_{\max}$. After truncation, the crossing symmetry equation (5.28) involves a finite number of unknown four-point structure constants,

$$X(N_{\max}) = \# \left( \left\{ D_i^{(c)} \right\}_{\Re(\Delta_i + \bar{\Delta}_i) \leq N_{\max}} \cup \left\{ D_{i'}^{(c')} \right\}_{\Re(\Delta_{i'} + \bar{\Delta}_{i'}) \leq N_{\max}} \right). \tag{5.30}$$

We then normalize one of these structure constants to be 1, and determine the rest by solving crossing symmetry for a number $E \geq X(N_{\max}) - 1$ of randomly chosen positions $Z^e = (z_1^e, z_2^e, z_3^e, z_4^e)$. We compute the averages and the relative deviations of the resulting structure constants over a number $A$ of random draws of the positions,

$$\bar{D}_i^{(c)} = \frac{1}{A} \sum_{a=1}^{A} D_i^{(c)}(Z_a^e) \quad , \quad \text{Deviation}\left( D_i^{(c)} \right) = \max_a \left( \frac{\left| D_i^{(c)}(Z_a^e) - \bar{D}_i^{(c)} \right|}{\max\left( \left| D_i^{(c)}(Z_a^e) \right|, \left| \bar{D}_i^{(c)} \right| \right)} \right). \tag{5.31}$$

If the crossing symmetry equation has a unique solution, the deviations should tend to zero as $N_{\max}$ increases, except for structure constants that are in fact zero.

**Results**

We focus on two specific examples of the crossing symmetry equations (5.28):

$$\sum_{i \in \mathcal{S}^{abab}_{\text{reduced}}} D_i^{abab} \mathcal{H}_i^{(s)}(z_k) = \sum_{i' \in \mathcal{S}^{aabb}_{\text{reduced}}} D_{i'}^{aabb} \mathcal{H}_{i'}^{(u)}(z_k) \,, \tag{5.32}$$

$$\sum_{i \in \mathcal{S}^{aaaa}_{\text{reduced}}} D_i^{aaaa} \left( \mathcal{H}_i^{(s)}(z_k) - \mathcal{H}_i^{(t)}(z_k) \right) = 0 \,. \tag{5.33}$$

In the first equation, we can use the normalization condition $D_{\langle 1,1 \rangle}^{aabb} = 1$, and determine the rest of the structure constants. In the second equation, we can then normalize the structure constants such that the relation (5.8) is obeyed. We find that both equations have unique solutions. For example, let us display how the deviations for $\mathcal{S}_{\text{reduced}}^{aabb}$ behave as the cutoff $N_{\text{max}}$ increases [9]:

$N_{\text{max}} = 16$

| $(r,s)$ | Deviation |
|---------|-----------|
| $(1,1)$ | $7.61 \times 10^{-12}$ |
| $(0,1/2)$ | $8.19 \times 10^{-12}$ |
| $(2,0)$ | $4.37 \times 10^{-11}$ |
| $(4,0)$ | $6.19 \times 10^{-8}$ |
| $(4,1/2)$ | $6.12 \times 10^{-8}$ |
| $(6,0)$ | $0.269$ |
| $(6,1/3)$ | $0.802$ |

$N_{\text{max}} = 24$

| $(r,s)$ | Deviation |
|---------|-----------|
| $(1,1)$ | $3.47 \times 10^{-18}$ |
| $(0,1/2)$ | $3.78 \times 10^{-18}$ |
| $(2,0)$ | $9.85 \times 10^{-18}$ |
| $(4,0)$ | $1.04 \times 10^{-14}$ |
| $(4,1/2)$ | $9.88 \times 10^{-15}$ |
| $(6,0)$ | $4.44 \times 10^{-8}$ |
| $(6,1/3)$ | $8.25 \times 10^{-8}$ |
| $(8,0)$ | $0.211$ |
| $(8,1/4)$ | $0.118$ |
| $(8,1/2)$ | $0.574$ |

$N_{\text{max}} = 32$

| $(r,s)$ | Deviation |
|---------|-----------|
| $(1,1)$ | $2.3 \times 10^{-24}$ |
| $(0,1/2)$ | $2.38 \times 10^{-24}$ |
| $(2,0)$ | $2.28 \times 10^{-24}$ |
| $(4,0)$ | $3.17 \times 10^{-21}$ |
| $(4,1/2)$ | $2.62 \times 10^{-21}$ |
| $(6,0)$ | $1.2 \times 10^{-14}$ |
| $(6,1/3)$ | $2.14 \times 10^{-14}$ |
| $(8,0)$ | $4.86 \times 10^{-7}$ |
| $(8,1/4)$ | $3.41 \times 10^{-7}$ |
| $(8,1/2)$ | $2.37 \times 10^{-7}$ |
| $(10,0)$ | $0.489$ |
| $(10,1/5)$ | $1.19$ |
| $(10,2/5)$ | $1.76$ |

$$(5.34)$$

Here and in our other numerical examples, we chose a generic value of the parameter $\beta = 0.8 + 0.1i$ i.e. $Q \simeq -0.121 + 1.725i$. The code runs in $O(10^3)$ seconds on a standard laptop. The deviation of $D_{\langle 1,1 \rangle}^{aabb}$ is nonzero because the code uses the normalization condition $D_{(0,\frac{1}{2})}^{abab} = 1$ rather than $D_{\langle 1,1 \rangle}^{aabb} = 1$.

In order to determine the structure constants $D_i^{abab}$, it may seem easier to focus on a crossing symmetry equation with fewer unknowns,

$$\sum_{i \in \mathcal{S}_{\text{reduced}}^{abab}} D_i^{abab} \left( \mathcal{H}_i^{(s)}(z_k) - (-1)^{\text{Spin}(i)} \mathcal{H}_i^{(t)}(z_k) \right) = 0 \,, \tag{5.35}$$

where the spin-dependent sign is due to the relation (5.6). However, while the deviation of $D_{(2,\frac{1}{2})}^{abab}$ does tend to zero as $N_{\text{max}}$ increases, the deviations of $D_{(r \geq 4, s)}^{abab}$ remain large. The intepretation is that our equation has a solution, which is however not unique. For the deviation of a structure constant to tend to zero, that structure constant must be nonvanishing in one solution only. We find an infinite series of subsets of the spectrum

$$(r_0 \in 2\mathbb{N}^*) \qquad \mathcal{S}_{\text{reduced}}^{r_0} = \left\{ V_{(r_0,0)}^N \right\} \cup \left\{ V_{(r,s)}^N \right\}_{\substack{r \geq r_0 \in 2\mathbb{N}^* \\ s \in \frac{1}{r}\mathbb{N}^* \cap [0,\frac{1}{2}]}} \subset \mathcal{S}_{\text{reduced}}^{abab} \,, \tag{5.36}$$

such that the crossing symmetry equation for $\mathcal{S}_{\text{reduced}}^{r_0}$ has a unique solution. These solutions provide a basis of solutions for the original crossing symmetry equation (5.35). Let us display

the deviations in the cases $r_0 = 2, 4, 6$ with $N_{\max} = 32$ [9]:

$r_0 = 2$

| $(r,s)$ | Deviation |
|---------|-----------|
| $(2,0)$ | $0$ |
| $(2,1/2)$ | $2.98 \times 10^{-28}$ |
| $(4,1/4)$ | $2.01 \times 10^{-23}$ |
| $(4,1/2)$ | $3.1 \times 10^{-24}$ |
| $(6,1/6)$ | $3.56 \times 10^{-17}$ |
| $(6,1/3)$ | $1.21 \times 10^{-16}$ |
| $(6,1/2)$ | $4.71 \times 10^{-16}$ |
| $(8,1/8)$ | $7.42 \times 10^{-5}$ |
| $(8,1/4)$ | $0.000244$ |
| $(8,3/8)$ | $0.000406$ |
| $(8,1/2)$ | $0.000243$ |

$r_0 = 4$

| $(r,s)$ | Deviation |
|---------|-----------|
| $(4,0)$ | $0$ |
| $(4,1/4)$ | $3.57 \times 10^{-25}$ |
| $(4,1/2)$ | $3.25 \times 10^{-25}$ |
| $(6,1/6)$ | $2.8 \times 10^{-15}$ |
| $(6,1/3)$ | $2.52 \times 10^{-15}$ |
| $(6,1/2)$ | $2.51 \times 10^{-15}$ |
| $(8,1/8)$ | $0.00143$ |
| $(8,1/4)$ | $0.0014$ |
| $(8,3/8)$ | $0.00137$ |
| $(8,1/2)$ | $0.00136$ |

$r_0 = 6$

| $(r,s)$ | Deviation |
|---------|-----------|
| $(6,0)$ | $0$ |
| $(6,1/6)$ | $1.76 \times 10^{-17}$ |
| $(6,1/3)$ | $4.62 \times 10^{-17}$ |
| $(6,1/2)$ | $6.39 \times 10^{-17}$ |
| $(8,1/8)$ | $0.105$ |
| $(8,1/4)$ | $0.105$ |
| $(8,3/8)$ | $0.105$ |
| $(8,1/2)$ | $0.105$ |

$$(5.37)$$

(For conciseness we do not display the deviations for the structure constants $D^{r_0}_{(10,s)}$, which are of order 1.)

**Comparison with analytic formulas**

Using a lattice regularization of the $Q$-state Potts model, analytic formulas for a few ratios of four-point structure constants have been conjectured [31](Section 5.2):

$$\frac{D^{aabb}_{(2,0)}}{D^{aaaa}_{(2,0)}} = \frac{1}{1-Q} \,, \tag{5.38}$$

$$\frac{D^{aabb}_{(4,0)}}{D^{aaaa}_{(4,0)}} = -\frac{Q^5 - 7Q^4 + 15Q^3 - 10Q^2 + 4Q - 2}{2(Q^2 - 3Q + 1)} \,, \tag{5.39}$$

$$\frac{D^{aabb}_{(4,\frac{1}{2})}}{D^{aaaa}_{(4,\frac{1}{2})}} = \frac{2-Q}{2} \,, \tag{5.40}$$

$$\frac{D^{abab}_{(2,0)}}{D^{aaaa}_{(2,0)}} = \frac{2-Q}{2} \,, \tag{5.41}$$

$$\frac{D^{abab}_{(4,0)}}{D^{aaaa}_{(4,0)}} = -\frac{1}{4}(Q^2 - 4Q + 2)(Q^2 - 3Q - 2) \,, \tag{5.42}$$

$$\frac{D^{abab}_{(4,\frac{1}{2})}}{D^{aaaa}_{(4,\frac{1}{2})}} = \frac{1}{4}(Q-1)(Q-4) \,. \tag{5.43}$$

We have checked these formulas to a high precision [9]. With our code, it is possible to look for analytic formulas for other ratios of structure constants. For instance, we have found the

formulas

$$\frac{D^{aabb}_{(6,\frac{1}{3})}}{D^{aaaa}_{(6,\frac{1}{3})}} = \frac{2-Q}{2} \,, \tag{5.44}$$

$$\frac{D^{abab}_{(6,\frac{1}{3})}}{D^{aaaa}_{(6,\frac{1}{3})}} = \frac{1}{4}\left(Q^5 - 9Q^4 + 27Q^3 - 28Q^2 + Q + 4\right) \,, \tag{5.45}$$

$$\frac{D^{aabb}_{(6,0)}}{D^{aaaa}_{(6,0)}} = \frac{2Q^8 - 26Q^7 + 134Q^6 - 348Q^5 + 479Q^4 - 337Q^3 + 112Q^2 - 23Q + 3}{(1 - 6Q + 5Q^2 - Q^3)(3Q^6 - 24Q^5 + 64Q^4 - 66Q^3 + 24Q^2 - 8Q + 3)} \,, \tag{5.46}$$

$$\frac{D^{abab}_{(6,0)}}{D^{aaaa}_{(6,0)}} = \frac{(2-Q)\left(Q^2 - 4Q + 1\right)\left(Q^6 - 9Q^5 + 30Q^4 - 40Q^3 + 13Q^2 + 4Q + 3\right)}{2\left(3Q^6 - 24Q^5 + 64Q^4 - 66Q^3 + 24Q^2 - 8Q + 3\right)} \,, \tag{5.47}$$

which hold at high precision.

### 5.4 The Delfino–Viti conjecture

Our semi-analytic bootstrap calculations give us access to the three-point structure constant $C_{(0,\frac{1}{2})(0,\frac{1}{2})(0,\frac{1}{2})} = \sqrt{D^{aaaa}_{(0,\frac{1}{2})}}$, which can be interpreted as the three-point connectivity [27]. Let us compare this with the conjectured exact expression of this connectivity.

Delfino and Viti's conjectured expression [32] is the three-point structure constant of Liouville theory with $c \leq 1$, times a combinatorial prefactor $\sqrt{2}$ whose lattice origin was elucidated in [33]:

$$C_{(0,\frac{1}{2})(0,\frac{1}{2})(0,\frac{1}{2})} = \sqrt{2} C^{c \leq 1 \text{ Liouville}}_{\Delta_{(0,\frac{1}{2})}, \Delta_{(0,\frac{1}{2})}, \Delta_{(0,\frac{1}{2})}} \,. \tag{5.48}$$

As far as we understand, Liouville theory appeared here not because it has any particular relation with the $Q$-state Potts model, but because it exists at generic central charges, has a diagonal field of dimension $\Delta_{(0,\frac{1}{2})}$, and is analytically solvable. Based on this information alone, we would a priori not expect the conjecture to be exactly true. And consistency with Monte-Carlo simulations of the $Q$-state Potts model [27, 34] only tests the conjecture to a relatively low precision.

However, it was recently observed that connectivities of the $Q$-state Potts model are related to correlation functions of the RSOS model [31]. In the critical limit, the latter model is described by analytically solvable CFTs of the type of Liouville theory and minimal models. This suggests that the Delfino–Viti conjecture may be exactly true. And this is what our numerical results confirm, with a precision of about 26 significant digits [9].

Let us emphasize that the $Q$-state Potts model is nevertheless not related to Liouville theory proper. The former has a discrete spectrum, the latter a continuous spectrum. The analytic expression for $C^{c \leq 1 \text{ Liouville}}_{\Delta_{(0,\frac{1}{2})}, \Delta_{(0,\frac{1}{2})}, \Delta_{(0,\frac{1}{2})}}$ is valid in Liouville theory for $c \leq 1$ only [22], while the three-point connectivity is valid under the much weaker condition $\Re c < 13$. The analytic expression for $C^{c \leq 1 \text{ Liouville}}_{\Delta_{(0,\frac{1}{2})}, \Delta_{(0,\frac{1}{2})}, \Delta_{(0,\frac{1}{2})}}$ is the unique solution of certain crossing symmetry equations, and should be considered a universal quantity, although it happened to be first discovered in the context of Liouville theory.

## 6 Conclusion and outlook

Let us summarize some of our results, and point out a few questions that arise.

**Logarithmizing CFT**

Using derivative fields, we found it relatively simple to derive logarithmic from non-logarithmic CFT objects: we have *logarithmized* Verma modules with zero or one null vector, as well as the associated correlation functions and conformal blocks. It would be interesting to understand this operation at a more formal level, in order to apply it to more complicated situations, including higher-dimensional CFT.

In two dimensions, *logarithmizing* Verma modules with null vectors leads to so-called staggered Virasoro modules [11]. For $\beta^2 \in \mathbb{Q}_{<0}$, there are infinitely many null vectors, and the resulting representations are relevant in CFTs such as critical percolation. Some of our results can be directly applied to these staggered Verma modules, as we saw in Section 3.1 when studying logarithmic couplings in a few examples. It would be interesting to study the applicability of our results for rational $\beta^2$ more systematically.

An alternative to *logarithmizing* non-logarithmic CFT at rational central charge would be to *rationalize* logarithmic CFT at generic central charge, by taking limits $\beta^2 \to \frac{p}{q}$. This would be straightforward if we wanted to numerically compute connectivities in critical percolation: we would just need to do it in the $Q$-state Potts model with a parameter $Q \approx 1$ i.e. $c \approx 0$. However, it would be more difficult to derive the space of states, structure constants and conformal blocks at $c = 0$: *rationalizing* is already known to be non-trivial in simpler, non-logarithmic CFTs [21].

If *logarithmizing* and *rationalizing* could be defined precisely, a natural question would be whether they commute.

**Crossing-symmetric four-point functions**

We have found that the crossing symmetry equations (5.32), (5.33) have unique solutions. This is a non-trivial result, which depends very sensitively on the structure of the spectrum, and on the correct computation of conformal blocks.

As always in the conformal bootstrap approach, the next question is to interepret the solutions, and determine to which CFT they belong. In the case of the Potts model, it is tricky to identify bootstrap solutions with four-point connectivities, because there are no high-precision results on connectivities. Based on Monte-Carlo calculations, four-point connectivities were found to agree with other solutions of crossing symmetry [30]. However, these other solutions turned out to fail more extensive numerical and analytic comparisons with the Potts model [27, 29], and to describe the RSOS model instead [31].

Let us summarize the argument for our bootstrap solutions to describe connectivities in the Potts model:

- The spectrum of the Potts model is known [25].

- More precisely, the spectrums of the connectivities in all channels are known [29].

- Crossing symmetry equations with these spectrums have unique solutions.

The bootstrap solutions of [30] featured only subsets of the expected spectrums, only in two out of three channels, and only for three out of four connectivities.

In the case of the particular crossing symmetry equation (5.35), we actually found an infinite-dimensional space of solutions, which includes the connectivity $P_{abab}$. This echos a speculation of [8] (Section 3.3), which predicted the existence of multiple solutions of crossing symmetry equations, based on the freedom to change the weights of non-contractible loops in the discretized model. Nevertheless, it is not yet clear what our extra solutions describe, or to which CFT they belong. There are not too many known solutions of crossing symmetry at generic central charge [35], so these solutions may be worth investigating.

**Towards a solution of the $Q$-state Potts model**

With our high-precision checks of analytic conjectures for structure constants or ratios thereof, we provided additional evidence that the $Q$-state Potts model may be analytically solvable. Of course, solving the model involves computing not just four-point connectivities, but also more general correlation functions. Connectivities have a few nice peculiarities, for instance their structure constants obey slightly stronger linear relations as we saw in Section 5.2, but this should not make an essential difference.

Meanwhile, the problem of numerically computing four-point connectivities to arbitrary precision is now effectively solved. After numerically determining four-point structure constants by solving crossing symmetry equations, we can use the structure constants for computing connectivities. An estimate of the precision of such computations is given by the lowest nonzero deviation of a structure constant, for example $O(10^{-24})$ for the last table of Eq. (5.34).

# Acknowledgements

We are grateful to Linnea Grans-Samuelsson, Yifei He, Jesper Jacobsen, and Hubert Saleur, for discussing their work on the $Q$-state Potts model. We are also indebted to them, as well as to Miguel Paulos and Jacopo Viti, for comments on the draft of this article. We would like to thank Riccardo Guida for help with speeding up Python code. We are grateful to Raoul Santachiara and Nina Javerzat for stimulating discussions. Many thanks to David Ridout for very stimulating exchanges on the draft of this article, which in particular led us to pay more attention to the logarithmic coupling.

We are grateful to the SciPost reviewers for their publicly available comments and suggestions.

This paper is partly a result of the ERC-SyG project, Recursive and Exact New Quantum Theory (ReNewQuantum) which received funding from the European Research Council (ERC) under the European Union's Horizon 2020 research and innovation programme under grant agreement No 810573.

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
