# Peer review of "Logarithmic CFT at generic central charge: from Liouville theory to the $Q$-state Potts model"

_SciPost Physics, doi:SciPost Phys. 10, 021 (2021)_

## Round 2 · Referee Report · Anonymous (Referee 1) · 2020-10-28

Report

This manuscript deals with logarithmic fields in 2d Conformal Field Theories (CFTs). The main point of the paper is to show that the logarithmic pairs (or multiplets) obtained generically in a CFT with a continuous spectrum by a differentiation procedure, can be used to construct some relevant conformal blocks for connectivity correlation functions in the Potts and O(n) models.

Sections 2 and 3 are devoted to generic CFTs with a continuous spectrum, where logarithmic fields can be defined by differentiating primary fields with respect to the conformal dimension. The construction is detailed in Section 2, and the two- and four-point correlators are studied in Section 3. In Section 4, a conjecture is proposed for the logarithmic structures appearing in the Potts and O(n) models, involving the logarithmic fields defined above. In Section 5, some ratios of amplitudes of four-point conformal blocks are determined analytically by the conformal bootstrap method. Then, the above conjecture is tested numerically on the four-point connectivity correlation functions of the cluster model associated to the Potts model. This is done by first combining the conformal blocks belonging to the same topological sectors as in Ref [7]; and then solving numerically the crossing symmetry constraints as in Ref [28].

The manuscript is correctly written, concise and structured. It presents some interesting proposals and results on a subject of interest for the Mathematical Physics community. However, some important issues should be mentioned:

  1. The construction of logarithmic fields with a differentiation procedure is presented as an original finding of the authors. However, this approach was already presented in: I.I. Kogan and A. Lewis, Origin of logarithmic operators in conformal field theories, Nucl. Phys. B509 (1998) 687-704. Considering the substantial overlap of Sections 2 and 3 with this reference, it is necessary to clarify what parts of the construction are indeed original, and what is taken from (or reproduces) previous results.

  2. The OPE arguments in the end of Sections 2.2 and 2.3 are completely similar to those introduced in the seminal paper on logarithmic CFTs: V. Gurarie, Logarithmic operators in conformal field theory, Nucl. Phys. B410 (1993), 535. Hence this reference, as well as other sources for the results in Sections 2 and 3, should be cited properly.

  3. The conjecture given in Section 4 introduces logarithmic fields constructed in a CFT with a continuous spectrum, into CFTs with discrete spectra. This comes with no justification, and is only tested numerically with the numerical solution of crossing symmetry constraints, as in Ref [28]. But it was demonstrated in Ref [27] that this numerical approach failed to detect a wrong choice of internal spectrum for the four-point connectivity function. To justify the use of this method for the exploration of finer structures of the spectrum, the authors should at least give a convincing explanation of the failure of this method on previous, easier questions.

The above points cast doubt on the originality and validity of the results reported in this manuscript. Hence, I cannot recommend its publication in SciPost Physics, unless all these points are properly addressed.

  • validity: -
  • significance: -
  • originality: -
  • clarity: -
  • formatting: -
  • grammar: -

Author:  Sylvain Ribault  on 2020-12-11  [id 1071]

(in reply to Report 1 on 2020-10-28)

In order to address this Report's points, here is what we changed in the next version of our submission. (Due to a technical error that we made, this list of changes is not displayed together with the resubmission.)

  • We added a last paragraph to the introductory text in Section 2, in order to explain what is original in this section.

  • We added a paragraph before Eq. (2.30), in order to acknowledge the origins of OPE techniques, and to indicate how we go beyond earlier work.

  • We added references to the early works [3] and [14].

  • We added a last paragraph to the introductory text in Section 3, in order to explain what is original in this section.

  • We swapped the second and third paragraphs of the Outlook, now called Conclusion and outlook. We rewrote what is now the second paragraph, in order to better argue that our bootstrap solutions are connectivities, and to compare with the claims in ref. [30] (previously ref. [28]).

---

## Round 2 · Referee Report · Anonymous (Referee 2) · 2020-11-6

Report

The paper is devoted to logarithmic structures in two-dimensional conformal field theory (CFT) and to their relevance for the study of the Q-state Potts model. The latter allows a generalisation to non-integer values of Q (random cluster model) and for Q\to 1 describes percolation, a paradigmatic example of geometrical criticality. The determination of cluster connectivities is a challenging problem of CFT, since they correspond to multi-point correlation functions of fields that are non-degenerate and cannot be computed using standard techniques. In recent years, however, Delfino and Viti argued the exact result for the three-point connectivity relating it to the Liouville CFT at central charge c<1.

This result led to the study of the four-point problem within the conformal bootstrap approach. A conjecture for the spectrum of conformal dimensions leads to an expansion of the four-point connectivities over conformal blocks. Crossing symmetry equations for the expansion coefficients are obtained imposing the independence of the result on the fusion channel. Crossing symmetry was successfully checked in Ref. [7], with a numerical accuracy limited by the appearance of a logarithmic conformal block, namely a block related to non-diagonal action of the dilatation operator. In order to progress on this point, the present paper starts with an investigation of logarithmic representations of the conformal algebra. Technically, the authors build logarithmic representations from derivatives of primary fields with respect to the conformal dimension, a trick that simplifies the computation of conformal blocks and the implementation of the single-valuedness of correlation functions. The derivative trick is natural in theories with a continuous spectrum of conformal dimensions, in particular the Liouville theory that the authors use for an illustration of their method. It can be argued, however, that the results for the conformal blocks formally hold also for the Q-state Potts model, which has a discrete spectrum. This expectation is supported by the authors through the semi-analytical study of the Potts crossing symmetry equations. In the first place, they show that the accuracy of the determination of the four-point structure constants obtained truncating Zamolodchikov’s recursive representation of the conformal blocks increases with the truncation level, so that arbitrary precision can in principle be obtained. As a second result, they extend the list of conjectured analytical formulas for ratios of four-point structure constants obtained in Ref. [29]. Finally, they evaluate the three-point connectivity from the suitable four-point channel and recover the Delfino-Viti exact value with an impressive accuracy of 26 significant digits.

The paper provides a further step forward in the study of the conformal bootstrap in two dimensions and of its application to the Q-state Potts model and the difficult problem of geometrical criticality. I recommend publication in SciPost.

---

## Round 2 · Referee Report · Anonymous (Referee 3) · 2020-11-11

Report

The article under review studies the appearance of logarithmic singularities in the correlation functions of two-dimensional conformal field theories, assuming only Virasoro symmetry, and applies the results to the critical $O(n)$ and $Q$-state Potts models. More precisely, they review and extend the well known approach to logarithmic structure given by "differentiating with respect to a parameter", in this case the conformal dimension. This is limited to generic highest-weight representations, so the results and applications are intended to describe logarithmic behaviour for generic values of $n$ and $Q$. Nevertheless, the authors detail several nontrivial checks of their work, all of which are of considerable interest.

The article is generally well written, though I have a few comments concerning the style that should be considered. It reviews a significant amount of background material and moreover presents results that will be of interest to a broad spectrum of the mathematical physics community. I therefore recommend publication with appropriate revisions. A few are suggested below. As I lack sufficient expertise in the analytic formalism employed for the applications, these are mostly confined to the logarithmic fundamentals of Sections 2 and 3.

Requested changes

  1. First, I have a mild issue with the authors regularly claiming that the parameter $\kappa$ introduced in (2.14) (or (2.21)) is a simpler and normalisation-independent version of the logarithmic coupling $\beta$ introduced in the uncited work arXiv:0708.0802 [hep-th]. I suppose that this is intended to justify its appearance. Simplicity is debatable, but the claim that it is normalisation-independent is clearly false, see (2.19) for one of several normalisation assumptions.

More fundamentally, one of the main theorems of [10] states that (the chiral versions of) the logarithmic representations considered by the authors have a moduli space isomorphic to $\mathbb{C}$. The logarithmic coupling is therefore interpreted as a coordinate on this moduli space. It is thus obvious that one cannot obtain a normalisation-independent coordinate.

I propose that the authors rephrase their claims throughout and perhaps add a discussion of some potential practical advantages of their $\kappa$. For example, their algorithm for computing $\kappa$ allows one to use any degree-$rs$ annihilation operator. The standard definition of $\beta$ however requires a specific choice, so this freedom should be beneficial in calculations.

  1. A related issue is the comparative and subjective language employed in several places which I would prefer to be less controversial. For example, the first sentence of the introduction claims that chiral studies of logarithmic CFT only focus on "fairly complicated cases". I doubt that many of the people doing that work would agree. A second example is the phrase "not a particularly exciting quantity" appearing before (3.26) which seems somehow demeaning to the authors of [17].

  2. Another mild issue is that the authors seem to sometimes cite articles which do not seem to be original sources. For example, a 2016 article [3] is cited in the introduction for the derivative trick in relation to logarithmic structure, despite the fact that this was common knowledge even in the 90s. Similarly, the ideas reviewed in Section 3 clearly owe a lot to the uncited work arXiv:hep-th/9303160. Finally, citing one of the authors for the global Ward identities seems unnecessary.

  3. The type of "Virasoro only" logarithmic structures that the authors focus on have been studied by many groups, though mostly in the chiral or boundary cases. The term "logarithmic minimal model" is often used here, so the authors may wish to add a remark about the relationship between their work and this literature.

  4. One common criticism of the "differentiate with respect to the conformal dimension" approach is that is doesn't clearly delineate whether the result of differentiating actually gives fields in the spectrum of the given CFT. For example, one can do this with the free boson but the resulting logarithmic representations are not held to give fields of the CFT as far as I am aware. I think it would help if the authors could address this, specifically in terms of the applications to the $O(n)$ and $Q$-state Potts models.

  5. There seems to have an implicit assumption throughout that the logarithmic representations introduced are inequivalent if their $\kappa$ parameters are different, see for example the sentence before (2.24). Maybe I missed it, but I didn't find such a statement, nor a justification. Perhaps the authors could explain why this must be true. It would be enough to demonstrate that $\kappa$ is always proportional to $\beta$, preferably with a nonzero proportionality constant.

  6. I had some difficulty understanding the relevance of (2.44), especially as it gets mentioned before (3.26) as being much more exciting than $\beta$ for a certain $c=0$ model. Could the authors please add some explanation around (2.44) as to what this parameter means in the context of rank-$3$ Jordan blocks. I think this would be a very valuable addition as I am not aware of any serious work addressing the analogue(s) of logarithmic couplings for higher-rank Jordan blocks (despite the examples that have been studied in the literature, beginning with the uncited arXiv:hep-th/0604097).

  7. Should "four-point" and "three-point" be swapped in the last sentence in the paragraph after (3.35)?

  8. The interesting discussion at the end of Section 3 complains that "logarithmic values of $P$ would anyway have measure zero", concluding that Liouville theory should thus not be regarded as logarithmic. However, it is perhaps worth pointing out that almost all the well known logarithmic CFTs have this type of behaviour. This is, for example, exploited in the conjectural "standard module formalism" for modularity and Verlinde fusion, introduced in [1].

  9. At the start of Section 4, there are a couple of surprising remarks concerning the existence of fields that somehow do not contribute to the torus partition function. Could the authors explain quickly why these fields should be considered at all? Do they, for example, contribute to the partition function of a boundary sector or a higher-genus surface? I have a similar issue with the claim that the $O(n)$ and $Q$-state Potts model torus partition functions are linear combinations of characters with coefficients that are not positive integers. Surely this indicates that one has the wrong set of characters (or is interpreting "partition function" incorrectly).

  • validity: -
  • significance: -
  • originality: -
  • clarity: -
  • formatting: -
  • grammar: -

Author:  Sylvain Ribault  on 2020-12-11  [id 1072]

(in reply to Report 3 on 2020-11-11)

In order to address this Report's points, here is what we changed in the next version of our submission. (Due to a technical error that we made, this list of changes is not displayed together with the resubmission.)

  1. and 6. We have rewritten much of Section 2.2 in order to make these points clearer. In particular, Eq. (2.29) makes normalization-independence manifest, and we showed that $\kappa$ characterizes our representations (without reference to other work). We do not understand the argument that we cannot define a normalization-independent coordinate on the moduli space. Sorting out this issue might require a few exchanges, and might be more efficiently done by email. We are not citing the suggested reference as we already have [6] by the same authors. Citing more references for the same material would only be of historical interest, but our subject is logarithmic CFT, not its history.

  2. We have rephrased the beginning of the introduction, and removed the "not particularly exciting" comment.

  3. We have followed the suggestions, and introduced ref. [3].

  4. Logarithmic minimal models have rational central charges, and their representations have more complicated structures. We gave some instances where our results agree with logarithmic minimal models in Section 3.1, but as we state in the conclusion, more work is needed.

  5. We use differentiation to build and study logarithmic representations, but this says nothing on whether they appear in a given CFT. We now make this point more explicitly in the introduction.

  6. See 1.

  7. The meaning of the parameter of rank 3 Jordan blocks is the same as for the rank 2 case. We concentrated on explaining the rank 2 case better.

  8. Yes, this was an error, now corrected.

  9. We have nothing to add on this interesting and subtle issue. Four-point functions and torus partition functions have rather different properties, so comparing the contributions of logarithmic representations in these two objects may not be very illuminating. It would probably be better to compare four-point functions of the $GL(1|1)$ WZW model with four-point functions of Liouville theory, but this is outside the scope of our paper.

  10. We have added a few precisions on the torus partition function. For a better understanding of these puzzling features, we refer to the cited literature.

---

## Round 3 · Referee Report · Anonymous (Referee 1) · 2021-1-7

Report

The points raised in my previous reports have been addressed, so I recommend this manuscript for publication in SciPost Physics.

---

## Round 3 · Referee Report · Anonymous (Referee 2) · 2021-1-11

Report

In the revised version the authors took into account in a detailed way referee comments concerning the logarithmic CFT methods, mostly about references to previous works. What makes the paper of considerable interest is the ability of the authors to develop those methods and to make them applicable to the challenging case of the conformal bootstrap for the Q-state Potts model and percolation. The difficulty of the problem also requires the combination of analytical and numerical methods. The authors manage to progress in both directions and to obtain new results. I confirm my recommendation for publication.

---

## Round 3 · Referee Report · Anonymous (Referee 4) · 2021-1-12

Report

The work is good for newcomers to the field. I think this version can be published.

---

## Round 3 · List of Changes

added clarifications and references following reviewers' suggestions, in particular in Section 2.2

---

## Editorial Decision

published